# Leave no child behind: Using data from 1.7 million children from 67 developing countries to measure inequality within and between groups of births and to identify left behind populations

**Antonio P. Ramos**[1,2]*, **Martin J. Flores**[3], **Robert E. Weiss**[1,2]

**1** Department of Biostatistics, Fielding School of Public Health, UCLA, Los Angeles, CA, United States of America, **2** California Center for Population Research, UCLA, Los Angeles, CA, United States of America, **3** Department of General Internal Medicine, UCLA David Geffen School of Medicine, Los Angeles, CA, United States of America

* TOMRAMOS@UCLA.EDU

**Data Availability Statement:** All relevant data are available on the Demographic and Health Surveys website and publicly accessible upon registering

## Abstract

### Background

Goal 3.2 from the Sustainable Development Goals (SDG) calls for reductions in national averages of Under-5 Mortality. However, it is well known that within countries these reductions can coexist with left behind populations that have mortality rates higher than national averages. To measure inequality in under-5 mortality and to identify left behind populations, mortality rates are often disaggregated by socioeconomic status within countries. While socioeconomic disparities are important, this approach does not quantify within group variability since births from the same socioeconomic group may have different mortality risks. This is the case because mortality risk depends on several risk factors and their interactions and births from the same socioeconomic group may have different risk factor combinations. Therefore mortality risk can be highly variable within socioeconomic groups. We develop a comprehensive approach using information from multiple risk factors simultaneously to measure inequality in mortality and to identify left behind populations.

### Methods

We use Demographic and Health Surveys (DHS) data on 1,691,039 births from 182 different surveys from 67 low and middle income countries, 51 of which had at least two surveys. We estimate mortality risk for each child in the data using a Bayesian hierarchical logistic regression model. We include commonly used risk factors for monitoring inequality in early life mortality for the SDG as well as their interactions. We quantify variability in mortality risk within and between socioeconomic groups and describe the highest risk sub-populations.

via the following URL: https://www.dhsprogram.com/data/new-user-registration.cfm.

**Funding:** The corresponding author, Dr. Antonio Pedro Ramos, has his worked funded by by the Eunice Kennedy Shriver National Institute of Child Health & Human Development of the National Institutes of Health under Award Number K99HD088727. The funders of the study had no role in study design, data collection, data analysis, data interpretation, or writing of the report. The corresponding author had full access to all the data in the study and had final responsibility for the decision to submit for publication.

**Competing interests:** The authors have declared that no competing interests exist.

## Findings

For all countries there is more variability in mortality within socioeconomic groups than between them. Within countries, socioeconomic membership usually explains less than 20% of the total variation in mortality risk. In contrast, country of birth explains 19% of the total variance in mortality risk. Targeting the 20% highest risk children based on our model better identifies under-5 deaths than targeting the 20% poorest. For all surveys, we report efficiency gains from 26% in Mali to 578% in Guyana. High risk births tend to be births from mothers who are in the lowest socioeconomic group, live in rural areas and/or have already experienced a prior death of a child.

## Interpretation

While important, differences in under-5 mortality across socioeconomic groups do not explain most of overall inequality in mortality risk because births from the same socioeconomic groups have different mortality risks. Similarly, policy makers can reach the highest risk children by targeting births based on several risk factors (socioeconomic status, residing in rural areas, having a previous death of a child and more) instead of using a single risk factor such as socioeconomic status. We suggest that researchers and policy makers monitor inequality in under-5 mortality using multiple risk factors simultaneously, quantifying inequality as a function of several risk factors to identify left behind populations in need of policy interventions and to help monitor progress toward the SDG.

## 1 Introduction

Goal 3.2 from the Sustainable Development Goals (SDG) requires reductions in under-5 mortality (http://www.un.org/sustainabledevelopment/health/). However, these reductions can co-exist with socioeconomic inequalities within countries where some groups have much higher mortality risk than others. [1] Studies have suggested that some of the Millennium Development Goals, which preceded the SDG, have not been achieved within many countries because of high levels of inequality. [2] Monitoring and reducing inequities in under-5 mortality requires the identification of births that are at highest risk of death such that policy interventions can target them. [3] The United Nations (UN) General Assembly Resolution 68/261, which highlights the Sustainable Development Indicators as a central framework for making progress on reducing early-life mortality, recommends that health indicators should be disaggregated, where relevant, by income, sex, age, and other characteristics. [4, 5] Disaggregation of inequality by several demographic groups has a clear policy implication: leave no one behind.

The literature that monitors progress towards SDG often quantifies gaps in either key health outcomes, such as neonatal or under-5 mortality, or in the coverage of health services, such as prenatal care or sanitation. Researchers and policy makers monitor progress toward SDG by evaluating mortality rates broken down by stratifiers, including wealth quintiles, rural/urban residence, maternal education, maternal age, gender of the child and geographic location (see https://www.equidade.org/indicators). [5] Even outside SDG monitoring, equity based strategies to reduce under-5 mortality usually measure gaps in average mortality rates between large groups of births, such as births from different socioeconomic groups within the

same country [6–10]. Studies have also documented significant under-5 mortality inequities across other demographic categories such as race, ethnicity, and geographic location [11–13].

Public health policies seeking to reduce inequality in early-life mortality often target births from an easily defined group with a high average mortality rates, usually the poorest. [9, 14–19] A recent meta-analysis shows that most targeted interventions aiming to improve maternal and child health often address economic disparities through various incentive schemes like conditional cash transfers and voucher schemes. [20] For example, Cash Transfer Programs (CTP), currently implemented in many low and middle income countries (LMIC), often improve infant and child health. [21, 22] In Burkina Faso, families enrolled in conditional cash transfer schemes were required to obtain quarterly child growth monitoring at local health clinics for all children under 60 months of age. [23] In India, the randomized controlled trial (RCT) *Lentils for Vaccines* targeted the poor, as do most RCTs that aim to increase vaccine uptake, good nutrition, or child health more generally [24].

One important assumption underlying these approaches to measure inequality and target populations is that most of the variability in mortality risk exists between groups of births, not within them. If that is the case, (a) comparing average mortality rates between groups provides us with a complete picture of the inequality in mortality risk faced by children in the population and (b) targeting the group with the highest average mortality risk will reach most high risk births in the population and reduce overall inequalities. However, if the grouping factors used to monitor inequality have high levels of within-group variation in mortality risk, then monitoring inequality based solely on between group comparisons will miss most of the variability in mortality risk and monitors will not be able to identify important left behind populations that require intervention. [7] Using data from India a recent study shows that most of the variation in mortality risk exists within groups, not between groups, and that program targeting based on poverty alone can be inefficient. [25] This makes sense as it is well known that multiple risk factors are associated with under-5 mortality risk.

In this paper we develop a novel framework to monitor disparities in mortality risk and to identify high risk subpopulations that cannot be identified otherwise. Our novel approach uses data from several demographic variables and a Bayesian hierarchical model to estimate mortality risk for each birth in our data set. We use these estimates to investigate within and between group variability across several commonly used demographic stratifiers that are used to monitor progress toward the SDG's and make international comparisons in inequality in under-5 mortality. We identify children with the highest mortality risk in the population and show how to construct a targetable group that contains more deaths than other targetable groups of the same size that are based on only one risk factor, such as poverty. We identify the groups at highest risk in each country to gain insight on their needs. Our methodology supports UN recommendations to disaggregate health indicators by demographic stratifiers to guide inequality monitoring so that countries can meet SDG targets with equity. We offer a more comprehensive approach that considers the effects of multiple risk factors and their interactions on mortality risk.

## 2 Methods

Births are the units of our analysis. We first estimate mortality risk for each child in our data and then we use these estimates as inputs in our subsequent equity analysis.

### 2.1 Data sources

The data used in this study comes from multiple Demographic and Health Surveys (DHS) (https://dhsprogram.com/). These are nationally representative surveys that have been

conducted in more than 100 low and middle income countries since 1984 [26, 27]. We analyze under-5 mortality and we exclude births that did not occur at least five years prior to the survey. We exclude all births that happen 10 years or more before the date of the survey to minimize measurement error and censoring issues. The final data set includes information on 1,691,039 births from a total of 182 different surveys from 67 countries, 51 of which had at least two surveys.

## 2.2 Estimating mortality risk

Mortality risk is a latent variable that must be estimated from data. Given our goal to improve inequality monitoring of the SDG, we base our estimation on predictors that are commonly used in studies that quantify progress toward SDG (https://www.equidade.org/indicators): maternal age, wealth, gender, year of birth, place of residence (urban/rural), maternal education in years.

The probability density functions (pdf) of the the original wealth index scores do not have a common range across countries. To make them more comparable across surveys we transform these pdf's into cumulative distribution functions (cdf). This approach gives wealth scores from different countries and surveys a common range, the unit interval (0,1) and makes the results interpretable in terms of relative wealth, a proxy for socioeconomic status within the countries. Details of the transformation are given in the appendix.

We also include three other variables that are available in DHS surveys and could aid inequality monitoring and targeting. Geographical locations are well known risk factors for mortality, as mortality risk tends to be geographically clustered. Using sampling clusters from DHS in our model allows us to capture unmeasured variables at the local level that were not otherwise recorded in the data. Further, geographic locations can potentially be targeted by policy makers. Similarly, we also construct a $0 - 1$ indicator variable for whether a child was born to a mother that had already experienced a death of a previous child. Prior death summarizes a number of risk factors at the maternal level that are not measured by existing variables. It is a forward looking variable because it only uses information on prior births to inform risk for the current birth. In particular, information on future siblings deaths are not used to predict past deaths and it is coded zero for a mother's first birth. It is also an actionable risk factor because policy makers can potentially target births from those mothers, as they are identifiable. Finally, we include birth order, coded as a continuous variable.

We estimate child mortality for each birth in our data as a function of these predictors and their interactions in a Bayesian hierarchical logistic regression model. We fit one model to the data from each survey. To avoid model misspecification and allow for all important interactions among the risk factors, we include all two-way, three-way, and four-way interaction terms for all covariates in the model. We include piecewise linear splines to capture non-linear trends in mortality as a function of the continuous variables. To aid in the estimation and avoid overfitting, we place increasingly restrictive priors on the variance parameters of the random effects for the higher order interaction terms, which shrink effects toward zero. We incorporate a location random effect to model differences in risk between births from different locations.

## 2.3 Equity analysis

We use estimates of the posterior distribution of mortality risk for each child in our data to feed our equity analysis. We use 1000 Markov Chain Monte Carlo (MCMC) samples from our model todo so. For the boxplots we use these samples to calculate the expected mortality risk for each child and then we plot these quantities.

We use box plots to display the within and between group variability in fitted mortality risk stratified by the DHS-assigned wealth quintile. We formally quantify how much of the variability in mortality risk is explained by the wealth quintiles using a Bayesian ANOVA, which allows us to get point and interval estimates of the $R^2$. Details of the ANOVA methods are given in the appendix.

Finally, we investigate whether using multiple risk factors simultaneously can help to identify high risk births that should be targeted by policy interventions. Using the last survey from each country, we compare how many actual deaths occur among the 20% highest risk births from our model versus the 20% poorest births based on the wealth CDF variable. Under the assumption that intervention has the same cost for each birth, we calculate the efficiency gain in targeting the highest risk births versus the poorest births by dividing the difference in mortality rates between highest risk births and poorest births by mortality rates among the poorest times 100. We thus define the efficiency gain as $\frac{(\text{HRDeaths}-\text{PoorDeaths})}{\text{PoorDeaths}} \times 100$, where "HRDeaths" is mortality among the 20% highest risk births and "PoorDeaths" is defined as mortality among the poorest 20% of births. For each survey, we compare births in the high risk group to births not in the high risk group based on the following covariates: wealth, maternal education, maternal age, place of residency (urban/rural), whether the birth was born to a mother who has experienced a prior death of another child. We compare lower and higher mortality risk groups by using either risk ratios for categorical risk factors or mean risk difference for continuous risk factors.

**2.3.1 Incorporating uncertainty in the equity analysis.** We use estimates of the posterior distribution of mortality risk for each child in our data to feed our equity analysis. We use 1000 Markov Chain Monte Carlo (MCMC) samples from our model to do so. For the boxplots we use these samples to calculate the expected mortality risk for each child and then we plot these quantities. For ANOVA and other tabulations, we calculate a quantity for each MCMC sample so that we have a distribution of these quantities that can be used to calculate posterior means and intervals. These also allow us to implement significant tests.

## 2.4 Role of the funding source

We acknowledge financial support from the Eunice Kennedy Shriver National Institute Of Child Health & Human Development of the National Institutes of Health under Award Number K99HD088727 and CCPR's Population Research Infrastructure Grant P2C from NICHD: P2C-HD041022. The sponsor of the study had no role in study design, data analysis, data collection, data interpretation, or writing of the report. The corresponding author had full access to all the data in the study; all authors had final responsibility for the decision to submit for publication.

# 3 Results

## 3.1 Mortality by wealth quintile in the raw data

All results use individual births as the unit of the analysis. Summaries of the Demographic and Health Surveys (DHS) are presented in Table 1. Each row presents data for one survey. From left to right, the columns in Table 1 are the number of births in each survey (N); the under-5 mortality rate (U5MR), defined as the fraction of births who die before age five, both overall and for each wealth quintile; and the proportion of deaths that occurred to the top 80% in wealth, which we call the non-poor deaths (NPD) fraction. If there is perfect equity in mortality across socioeconomic groups, then the NPD would be exactly 80%. If the poorest 20% contain more than their share of deaths, then the NPD would be lower than 80%. Under-5

**Table 1.** N is the survey sample size used in our analysis. U5MR is the under-5 mortality rates by age five for each survey. Non-poor deaths (NPD) is the fraction of deaths from the top 80% wealth quintile. DRC is Democratic Republic of Congo, DR is Dominican Republic, and CAR is Central African Republic. The first quintile is the poorest births and the fifth quantile is the richest births.

| Country (Survey Year) | N | U5MR | U5MR by Wealth Quintile | | | | | NPD |
|---|---|---|---|---|---|---|---|---|
| | | | First | Second | Third | Fourth | Fifth | |
| Albania (2009) | 2,481 | 0.028 | 0.049 | 0.022 | 0.030 | 0.009 | 0.023 | 0.551 |
| Angola (2011) | 5,812 | 0.109 | 0.149 | 0.105 | 0.127 | 0.103 | 0.088 | 0.819 |
| Armenia (2000) | 2,602 | 0.057 | 0.058 | 0.059 | 0.056 | 0.066 | 0.043 | 0.736 |
| Armenia (2010) | 1,545 | 0.028 | 0.028 | 0.052 | 0.023 | 0.013 | 0.022 | 0.814 |
| Azerbaijan (2006) | 2,739 | 0.063 | 0.071 | 0.072 | 0.056 | 0.052 | 0.055 | 0.680 |
| Bangladesh (2000) | 9,061 | 0.127 | 0.164 | 0.158 | 0.110 | 0.109 | 0.082 | 0.717 |
| Bangladesh (2004) | 7,261 | 0.101 | 0.120 | 0.112 | 0.094 | 0.091 | 0.080 | 0.725 |
| Bangladesh (2007) | 6,929 | 0.083 | 0.100 | 0.105 | 0.091 | 0.073 | 0.046 | 0.733 |
| Bangladesh (2014) | 14,512 | 0.061 | 0.080 | 0.067 | 0.058 | 0.057 | 0.036 | 0.687 |
| Benin (1996) | 5,386 | 0.200 | 0.224 | 0.213 | 0.212 | 0.196 | 0.116 | 0.722 |
| Benin (2001) | 5,691 | 0.170 | 0.211 | 0.183 | 0.168 | 0.141 | 0.109 | 0.694 |
| Benin (2006) | 16,984 | 0.152 | 0.169 | 0.165 | 0.161 | 0.143 | 0.091 | 0.728 |
| Benin (2012) | 12,904 | 0.084 | 0.093 | 0.100 | 0.087 | 0.074 | 0.043 | 0.723 |
| Bolivia (1998) | 9,334 | 0.117 | 0.161 | 0.125 | 0.119 | 0.062 | 0.044 | 0.574 |
| Bolivia (2004) | 10,546 | 0.103 | 0.128 | 0.126 | 0.103 | 0.068 | 0.046 | 0.688 |
| Bolivia (2008) | 10,048 | 0.080 | 0.112 | 0.087 | 0.074 | 0.060 | 0.029 | 0.597 |
| Brazil (1996) | 6,023 | 0.071 | 0.113 | 0.067 | 0.045 | 0.037 | 0.036 | 0.477 |
| Burkina Faso (1993) | 5,514 | 0.206 | 0.206 | 0.253 | 0.236 | 0.221 | 0.157 | 0.850 |
| Burkina Faso (1999) | 5,702 | 0.230 | 0.250 | 0.249 | 0.251 | 0.249 | 0.152 | 0.751 |
| Burkina Faso (2003) | 12,060 | 0.200 | 0.201 | 0.227 | 0.204 | 0.208 | 0.144 | 0.804 |
| Burkina Faso (2010) | 16,759 | 0.164 | 0.186 | 0.186 | 0.162 | 0.157 | 0.110 | 0.756 |
| Burundi (2011) | 6,016 | 0.137 | 0.170 | 0.163 | 0.152 | 0.136 | 0.074 | 0.761 |
| Cambodia (2000) | 12,071 | 0.131 | 0.171 | 0.144 | 0.120 | 0.116 | 0.072 | 0.646 |
| Cambodia (2011) | 7,258 | 0.081 | 0.113 | 0.104 | 0.084 | 0.050 | 0.038 | 0.633 |
| Cambodia (2014) | 8,272 | 0.060 | 0.093 | 0.073 | 0.051 | 0.041 | 0.029 | 0.611 |
| Cameroon (1991) | 3,140 | 0.149 | 0.210 | 0.204 | 0.146 | 0.131 | 0.088 | 0.771 |
| Cameroon (1998) | 4,080 | 0.145 | 0.212 | 0.176 | 0.145 | 0.101 | 0.096 | 0.662 |
| Cameroon (2004) | 7,535 | 0.157 | 0.207 | 0.181 | 0.155 | 0.102 | 0.090 | 0.645 |
| Cameroon (2011) | 10,812 | 0.133 | 0.188 | 0.148 | 0.126 | 0.095 | 0.076 | 0.676 |
| CAR (1995) | 4,429 | 0.166 | 0.204 | 0.181 | 0.167 | 0.166 | 0.093 | 0.692 |
| Chad (1997) | 6,941 | 0.201 | 0.173 | 0.230 | 0.227 | 0.223 | 0.167 | 0.854 |
| Chad (2004) | 6,260 | 0.201 | 0.191 | 0.215 | 0.231 | 0.217 | 0.178 | 0.822 |
| Chad (2015) | 18,985 | 0.144 | 0.160 | 0.160 | 0.136 | 0.132 | 0.135 | 0.798 |
| Colombia (1990) | 4,087 | 0.041 | 0.069 | 0.055 | 0.034 | 0.032 | 0.025 | 0.754 |
| Colombia (1995) | 5,041 | 0.040 | 0.053 | 0.041 | 0.029 | 0.042 | 0.026 | 0.655 |
| Colombia (2005) | 15,630 | 0.032 | 0.047 | 0.032 | 0.026 | 0.020 | 0.021 | 0.598 |
| Comoros (1996) | 2,208 | 0.116 | 0.132 | 0.139 | 0.108 | 0.094 | 0.091 | 0.715 |
| Comoros (2012) | 3,390 | 0.050 | 0.051 | 0.054 | 0.052 | 0.055 | 0.035 | 0.725 |
| DRC (2005) | 4,419 | 0.134 | 0.157 | 0.141 | 0.137 | 0.143 | 0.081 | 0.745 |
| DRC (2007) | 7,971 | 0.172 | 0.207 | 0.195 | 0.180 | 0.155 | 0.107 | 0.734 |
| DRC (2012) | 7,597 | 0.097 | 0.105 | 0.106 | 0.082 | 0.066 | 0.071 | 0.501 |
| DRC (2014) | 15,132 | 0.125 | 0.137 | 0.137 | 0.124 | 0.128 | 0.077 | 0.717 |
| Côte d'Ivoire (1999) | 2,757 | 0.158 | 0.195 | 0.172 | 0.189 | 0.136 | 0.110 | 0.789 |
| Côte d'Ivoire (2005) | 3,812 | 0.127 | 0.149 | 0.127 | 0.125 | 0.115 | 0.097 | 0.673 |
| Côte d'Ivoire (2012) | 7,224 | 0.140 | 0.145 | 0.145 | 0.170 | 0.124 | 0.087 | 0.762 |

(*Continued*)

**Table 1.** (Continued)

| Country (Survey Year) | N | U5MR | U5MR by Wealth Quintile | | | | | NPD |
|---|---|---|---|---|---|---|---|---|
| | | | First | Second | Third | Fourth | Fifth | |
| Dominican Republic (1999) | 3,250 | 0.070 | 0.093 | 0.074 | 0.071 | 0.049 | 0.019 | 0.575 |
| Dominican Republic (2002) | 12,941 | 0.049 | 0.071 | 0.045 | 0.039 | 0.039 | 0.019 | 0.541 |
| Dominican Republic (2007) | 13,945 | 0.037 | 0.047 | 0.037 | 0.032 | 0.025 | 0.028 | 0.558 |
| Dominican Republic (2013) | 4,782 | 0.042 | 0.057 | 0.042 | 0.032 | 0.024 | 0.023 | 0.505 |
| Egypt (1996) | 12,791 | 0.110 | 0.158 | 0.133 | 0.107 | 0.070 | 0.038 | 0.605 |
| Egypt (2003) | 11,850 | 0.070 | 0.099 | 0.079 | 0.067 | 0.047 | 0.036 | 0.611 |
| Egypt (2008) | 11,394 | 0.039 | 0.061 | 0.035 | 0.035 | 0.026 | 0.025 | 0.592 |
| Egypt (2014) | 14,486 | 0.035 | 0.051 | 0.043 | 0.033 | 0.028 | 0.021 | 0.700 |
| Eswatini (2007) | 2,421 | 0.102 | 0.118 | 0.108 | 0.097 | 0.102 | 0.091 | 0.782 |
| Ethiopia (1997) | 12,984 | 0.141 | 0.134 | 0.168 | 0.153 | 0.158 | 0.104 | 0.743 |
| Ethiopia (2003) | 13,218 | 0.129 | 0.149 | 0.132 | 0.132 | 0.135 | 0.084 | 0.636 |
| Gabon (2001) | 3,783 | 0.093 | 0.095 | 0.117 | 0.099 | 0.083 | 0.040 | 0.685 |
| Gabon (2012) | 5,149 | 0.070 | 0.082 | 0.073 | 0.061 | 0.047 | 0.035 | 0.453 |
| Ghana (1994) | 3,281 | 0.147 | 0.181 | 0.188 | 0.155 | 0.114 | 0.078 | 0.751 |
| Ghana (1999) | 3,226 | 0.126 | 0.156 | 0.142 | 0.126 | 0.103 | 0.048 | 0.565 |
| Ghana (2003) | 4,134 | 0.127 | 0.155 | 0.120 | 0.125 | 0.112 | 0.088 | 0.603 |
| Ghana (2008) | 3,258 | 0.096 | 0.114 | 0.093 | 0.105 | 0.076 | 0.068 | 0.620 |
| Ghana (2014) | 6,370 | 0.084 | 0.107 | 0.077 | 0.067 | 0.062 | 0.080 | 0.570 |
| Guatemala (1999) | 7,083 | 0.078 | 0.085 | 0.087 | 0.081 | 0.065 | 0.034 | 0.637 |
| Guatemala (2015) | 11,719 | 0.041 | 0.057 | 0.043 | 0.038 | 0.030 | 0.021 | 0.623 |
| Guinea (1999) | 6,867 | 0.195 | 0.235 | 0.218 | 0.196 | 0.182 | 0.128 | 0.721 |
| Guinea (2005) | 7,807 | 0.201 | 0.219 | 0.230 | 0.220 | 0.172 | 0.125 | 0.741 |
| Guinea (2012) | 8,010 | 0.143 | 0.180 | 0.151 | 0.152 | 0.122 | 0.073 | 0.684 |
| Guyana (2005) | 1,268 | 0.046 | 0.030 | 0.052 | 0.029 | 0.038 | 0.088 | 0.828 |
| Guyana (2009) | 2,464 | 0.037 | 0.027 | 0.042 | 0.030 | 0.065 | 0.036 | 0.700 |
| Haiti (1995) | 3,020 | 0.157 | 0.194 | 0.182 | 0.147 | 0.139 | 0.100 | 0.722 |
| Haiti (2000) | 7,063 | 0.152 | 0.185 | 0.140 | 0.151 | 0.137 | 0.114 | 0.655 |
| Haiti (2006) | 5,907 | 0.107 | 0.130 | 0.115 | 0.098 | 0.098 | 0.067 | 0.655 |
| Haiti (2012) | 6,944 | 0.096 | 0.101 | 0.102 | 0.102 | 0.091 | 0.065 | 0.691 |
| Honduras (2006) | 12,380 | 0.045 | 0.055 | 0.052 | 0.041 | 0.029 | 0.022 | 0.606 |
| Honduras (2012) | 10,065 | 0.031 | 0.041 | 0.027 | 0.028 | 0.025 | 0.020 | 0.553 |
| India (1993) | 65,681 | 0.113 | 0.162 | 0.157 | 0.115 | 0.085 | 0.055 | 0.723 |
| India (2000) | 53,079 | 0.099 | 0.146 | 0.122 | 0.104 | 0.070 | 0.045 | 0.686 |
| India (2006) | 59,240 | 0.080 | 0.128 | 0.099 | 0.080 | 0.061 | 0.037 | 0.699 |
| Indonesia (1997) | 23,155 | 0.085 | 0.111 | 0.101 | 0.084 | 0.060 | 0.031 | 0.565 |
| Indonesia (2003) | 16,049 | 0.064 | 0.091 | 0.068 | 0.056 | 0.043 | 0.027 | 0.515 |
| Indonesia (2007) | 20,592 | 0.067 | 0.100 | 0.072 | 0.054 | 0.042 | 0.034 | 0.529 |
| Indonesia (2012) | 19,788 | 0.054 | 0.087 | 0.057 | 0.038 | 0.037 | 0.019 | 0.490 |
| Jordan (1990) | 9,308 | 0.046 | 0.061 | 0.056 | 0.043 | 0.038 | 0.034 | 0.796 |
| Jordan (1997) | 6,408 | 0.036 | 0.046 | 0.040 | 0.036 | 0.026 | 0.029 | 0.707 |
| Jordan (2002) | 7,098 | 0.037 | 0.040 | 0.041 | 0.037 | 0.031 | 0.029 | 0.708 |
| Jordan (2009) | 13,691 | 0.029 | 0.035 | 0.022 | 0.028 | 0.029 | 0.026 | 0.611 |
| Jordan (2012) | 11,205 | 0.024 | 0.029 | 0.023 | 0.021 | 0.024 | 0.015 | 0.670 |
| Kazakhstan (1999) | 2,651 | 0.057 | 0.069 | 0.062 | 0.067 | 0.052 | 0.038 | 0.762 |
| Kenya (1993) | 6,514 | 0.097 | 0.138 | 0.129 | 0.078 | 0.067 | 0.060 | 0.681 |
| Kenya (1998) | 5,789 | 0.104 | 0.140 | 0.119 | 0.104 | 0.076 | 0.058 | 0.668 |

(*Continued*)

**Table 1.** (Continued)

| Country (Survey Year) | N | U5MR | U5MR by Wealth Quintile | | | | | NPD |
|---|---|---|---|---|---|---|---|---|
| | | | First | Second | Third | Fourth | Fifth | |
| Kenya (2009) | 5,412 | 0.095 | 0.103 | 0.106 | 0.098 | 0.074 | 0.084 | 0.686 |
| Kenya (2014) | 23,924 | 0.055 | 0.053 | 0.066 | 0.054 | 0.053 | 0.044 | 0.674 |
| Kyrgyzstan (1997) | 2,400 | 0.074 | 0.094 | 0.092 | 0.079 | 0.051 | 0.043 | 0.669 |
| Kyrgyzstan (2012) | 3,705 | 0.036 | 0.031 | 0.037 | 0.048 | 0.032 | 0.031 | 0.799 |
| Lesotho (2005) | 3,115 | 0.093 | 0.113 | 0.107 | 0.090 | 0.075 | 0.077 | 0.746 |
| Lesotho (2010) | 3,107 | 0.087 | 0.077 | 0.095 | 0.098 | 0.090 | 0.079 | 0.737 |
| Lesotho (2014) | 3,250 | 0.100 | 0.080 | 0.103 | 0.117 | 0.120 | 0.087 | 0.791 |
| Liberia (2009) | 6,871 | 0.173 | 0.195 | 0.176 | 0.158 | 0.173 | 0.149 | 0.713 |
| Liberia (2013) | 8,220 | 0.132 | 0.147 | 0.131 | 0.123 | 0.108 | 0.126 | 0.618 |
| Madagascar (1997) | 5,960 | 0.165 | 0.208 | 0.186 | 0.178 | 0.137 | 0.098 | 0.675 |
| Madagascar (2004) | 5,268 | 0.106 | 0.163 | 0.142 | 0.114 | 0.095 | 0.058 | 0.699 |
| Madagascar (2009) | 12,686 | 0.087 | 0.111 | 0.098 | 0.093 | 0.070 | 0.045 | 0.651 |
| Malawi (1992) | 4,746 | 0.231 | 0.273 | 0.242 | 0.259 | 0.256 | 0.154 | 0.799 |
| Malawi (2005) | 9,663 | 0.180 | 0.216 | 0.192 | 0.193 | 0.167 | 0.124 | 0.777 |
| Malawi (2010) | 20,677 | 0.129 | 0.145 | 0.136 | 0.133 | 0.115 | 0.110 | 0.748 |
| Malawi (2016) | 16,793 | 0.079 | 0.094 | 0.082 | 0.088 | 0.076 | 0.053 | 0.756 |
| Mali (1996) | 9,960 | 0.259 | 0.310 | 0.292 | 0.262 | 0.238 | 0.175 | 0.757 |
| Mali (2001) | 13,031 | 0.257 | 0.264 | 0.271 | 0.287 | 0.271 | 0.148 | 0.776 |
| Mali (2006) | 15,201 | 0.222 | 0.248 | 0.261 | 0.229 | 0.210 | 0.134 | 0.773 |
| Mali (2013) | 9,249 | 0.113 | 0.120 | 0.140 | 0.130 | 0.108 | 0.063 | 0.779 |
| Moldova (2005) | 1,744 | 0.033 | 0.036 | 0.031 | 0.044 | 0.036 | 0.018 | 0.789 |
| Morocco (1992) | 5,422 | 0.088 | 0.110 | 0.094 | 0.092 | 0.074 | 0.050 | 0.695 |
| Morocco (2004) | 6,493 | 0.061 | 0.085 | 0.069 | 0.048 | 0.046 | 0.027 | 0.602 |
| Mozambique (1997) | 6,834 | 0.200 | 0.262 | 0.213 | 0.210 | 0.183 | 0.120 | 0.674 |
| Mozambique (2004) | 8,942 | 0.195 | 0.229 | 0.222 | 0.227 | 0.168 | 0.115 | 0.716 |
| Mozambique (2011) | 10,379 | 0.112 | 0.137 | 0.112 | 0.126 | 0.100 | 0.093 | 0.783 |
| Namibia (1992) | 3,692 | 0.109 | 0.137 | 0.100 | 0.103 | 0.120 | 0.079 | 0.718 |
| Namibia (2000) | 4,354 | 0.063 | 0.073 | 0.090 | 0.072 | 0.058 | 0.033 | 0.778 |
| Namibia (2007) | 4,668 | 0.069 | 0.097 | 0.078 | 0.064 | 0.062 | 0.032 | 0.703 |
| Namibia (2013) | 4,691 | 0.058 | 0.065 | 0.074 | 0.060 | 0.056 | 0.023 | 0.745 |
| Nicaragua (1998) | 8,665 | 0.062 | 0.067 | 0.070 | 0.060 | 0.054 | 0.041 | 0.661 |
| Nicaragua (2001) | 9,008 | 0.049 | 0.063 | 0.053 | 0.048 | 0.036 | 0.018 | 0.600 |
| Niger (1998) | 7,644 | 0.306 | 0.294 | 0.376 | 0.356 | 0.329 | 0.194 | 0.823 |
| Niger (2006) | 9,820 | 0.206 | 0.189 | 0.237 | 0.248 | 0.227 | 0.151 | 0.812 |
| Niger (2012) | 13,573 | 0.151 | 0.153 | 0.175 | 0.175 | 0.162 | 0.099 | 0.805 |
| Nigeria (1990) | 8,696 | 0.190 | 0.247 | 0.243 | 0.213 | 0.165 | 0.105 | 0.729 |
| Nigeria (2003) | 5,848 | 0.221 | 0.246 | 0.291 | 0.213 | 0.201 | 0.092 | 0.721 |
| Nigeria (2008) | 30,182 | 0.185 | 0.224 | 0.226 | 0.169 | 0.137 | 0.091 | 0.657 |
| Nigeria (2013) | 34,186 | 0.158 | 0.204 | 0.202 | 0.146 | 0.109 | 0.085 | 0.685 |
| Pakistan (1991) | 8,356 | 0.110 | 0.109 | 0.140 | 0.128 | 0.110 | 0.074 | 0.864 |
| Pakistan (2007) | 9,531 | 0.089 | 0.112 | 0.097 | 0.076 | 0.085 | 0.060 | 0.698 |
| Pakistan (2013) | 11,854 | 0.093 | 0.122 | 0.099 | 0.091 | 0.082 | 0.057 | 0.673 |
| Paraguay (1990) | 4,375 | 0.053 | 0.069 | 0.055 | 0.054 | 0.045 | 0.018 | 0.597 |
| Peru (1992) | 9,085 | 0.112 | 0.155 | 0.133 | 0.083 | 0.055 | 0.035 | 0.553 |
| Peru (1996) | 19,554 | 0.088 | 0.121 | 0.097 | 0.067 | 0.058 | 0.026 | 0.527 |
| Peru (2000) | 17,334 | 0.081 | 0.112 | 0.094 | 0.060 | 0.037 | 0.016 | 0.536 |

(*Continued*)

**Table 1.** (Continued)

| Country (Survey Year) | N | U5MR | U5MR by Wealth Quintile | | | | | NPD |
|---|---|---|---|---|---|---|---|---|
| | | | First | Second | Third | Fourth | Fifth | |
| Peru (2008) | 13,739 | 0.040 | 0.063 | 0.047 | 0.037 | 0.025 | 0.019 | 0.720 |
| Peru (2012) | 31,443 | 0.033 | 0.046 | 0.035 | 0.026 | 0.020 | 0.013 | 0.544 |
| Philippines (1993) | 9,340 | 0.075 | 0.101 | 0.088 | 0.068 | 0.038 | 0.052 | 0.625 |
| Philippines (1998) | 8,361 | 0.065 | 0.091 | 0.070 | 0.052 | 0.039 | 0.031 | 0.530 |
| Philippines (2003) | 7,863 | 0.045 | 0.073 | 0.048 | 0.033 | 0.020 | 0.023 | 0.526 |
| Philippines (2008) | 7,480 | 0.044 | 0.066 | 0.043 | 0.030 | 0.032 | 0.024 | 0.535 |
| Philippines (2013) | 8,159 | 0.033 | 0.051 | 0.032 | 0.025 | 0.018 | 0.017 | 0.485 |
| Rwanda (1992) | 6,071 | 0.174 | 0.165 | 0.218 | 0.155 | 0.211 | 0.134 | 0.795 |
| Rwanda (2005) | 9,139 | 0.202 | 0.223 | 0.224 | 0.200 | 0.224 | 0.132 | 0.744 |
| Rwanda (2008) | 4,865 | 0.149 | 0.176 | 0.166 | 0.159 | 0.159 | 0.087 | 0.824 |
| Rwanda (2015) | 8,096 | 0.071 | 0.082 | 0.082 | 0.077 | 0.068 | 0.040 | 0.731 |
| Sao Tome and Principe (2009) | 1,685 | 0.081 | 0.087 | 0.076 | 0.082 | 0.106 | 0.034 | 0.728 |
| Senegal (1997) | 7,311 | 0.157 | 0.189 | 0.192 | 0.165 | 0.109 | 0.076 | 0.706 |
| Senegal (2005) | 10,284 | 0.162 | 0.210 | 0.186 | 0.158 | 0.100 | 0.079 | 0.677 |
| Senegal (2009) | 13,229 | 0.124 | 0.154 | 0.135 | 0.107 | 0.063 | 0.067 | 0.575 |
| Senegal (2015) | 12,606 | 0.084 | 0.110 | 0.089 | 0.075 | 0.054 | 0.046 | 0.596 |
| Sierra Leone (2008) | 6,413 | 0.179 | 0.214 | 0.184 | 0.163 | 0.173 | 0.155 | 0.739 |
| Sierra Leone (2013) | 13,981 | 0.187 | 0.206 | 0.197 | 0.192 | 0.179 | 0.142 | 0.746 |
| South Africa (1998) | 5,564 | 0.057 | 0.085 | 0.073 | 0.048 | 0.031 | 0.022 | 0.610 |
| Tanzania (1999) | 6,715 | 0.150 | 0.159 | 0.167 | 0.167 | 0.169 | 0.098 | 0.764 |
| Tanzania (2005) | 7,200 | 0.143 | 0.166 | 0.158 | 0.160 | 0.124 | 0.101 | 0.755 |
| Tanzania (2010) | 11,262 | 0.101 | 0.126 | 0.110 | 0.098 | 0.092 | 0.071 | 0.737 |
| Tanzania (2016) | 8,745 | 0.079 | 0.085 | 0.081 | 0.076 | 0.084 | 0.062 | 0.755 |
| Timor-Leste (2010) | 9,499 | 0.089 | 0.096 | 0.102 | 0.095 | 0.091 | 0.059 | 0.758 |
| Togo (1998) | 7,211 | 0.155 | 0.174 | 0.181 | 0.159 | 0.119 | 0.102 | 0.720 |
| Togo (2014) | 6,901 | 0.109 | 0.131 | 0.122 | 0.112 | 0.084 | 0.045 | 0.588 |
| Turkey (1993) | 4,998 | 0.090 | 0.144 | 0.095 | 0.087 | 0.073 | 0.030 | 0.639 |
| Turkey (1998) | 4,162 | 0.064 | 0.096 | 0.065 | 0.058 | 0.045 | 0.033 | 0.615 |
| Turkey (2004) | 4,765 | 0.058 | 0.087 | 0.065 | 0.051 | 0.034 | 0.031 | 0.587 |
| Uganda (1995) | 6,244 | 0.159 | 0.199 | 0.183 | 0.158 | 0.163 | 0.114 | 0.778 |
| Uganda (2001) | 5,933 | 0.154 | 0.192 | 0.194 | 0.170 | 0.136 | 0.102 | 0.784 |
| Uganda (2010) | 5,912 | 0.142 | 0.168 | 0.149 | 0.138 | 0.134 | 0.104 | 0.690 |
| Uganda (2011) | 7,852 | 0.117 | 0.137 | 0.137 | 0.110 | 0.112 | 0.080 | 0.684 |
| Ukraine (2007) | 1,494 | 0.021 | 0.021 | 0.015 | 0.021 | 0.041 | 0.011 | 0.806 |
| Uzbekistan (1996) | 2,656 | 0.054 | 0.064 | 0.039 | 0.054 | 0.065 | 0.049 | 0.776 |
| Vietnam (2002) | 4,060 | 0.039 | 0.055 | 0.045 | 0.031 | 0.030 | 0.023 | 0.643 |
| Zambia (1997) | 5,614 | 0.192 | 0.214 | 0.226 | 0.192 | 0.169 | 0.126 | 0.660 |
| Zambia (2002) | 6,027 | 0.171 | 0.204 | 0.188 | 0.196 | 0.142 | 0.084 | 0.722 |
| Zambia (2007) | 5,808 | 0.147 | 0.125 | 0.171 | 0.172 | 0.142 | 0.102 | 0.821 |
| Zambia (2014) | 12,324 | 0.088 | 0.109 | 0.091 | 0.087 | 0.069 | 0.072 | 0.728 |
| Zimbabwe (1994) | 4,622 | 0.066 | 0.073 | 0.084 | 0.050 | 0.073 | 0.045 | 0.702 |
| Zimbabwe (1999) | 3,713 | 0.078 | 0.085 | 0.087 | 0.081 | 0.081 | 0.043 | 0.697 |
| Zimbabwe (2006) | 4,357 | 0.062 | 0.064 | 0.071 | 0.069 | 0.055 | 0.047 | 0.748 |
| Zimbabwe (2011) | 4,374 | 0.067 | 0.075 | 0.075 | 0.074 | 0.052 | 0.057 | 0.718 |
| Zimbabwe (2015) | 5,726 | 0.093 | 0.118 | 0.102 | 0.103 | 0.084 | 0.062 | 0.726 |

mortality rates are generally higher for the poorest wealth quintiles, reflecting a socioeconomic gradient in mortality. Some countries, such as Egypt, exhibit a consistent decrease in mortality with increasing wealth quintile. In a few countries, mortality increases from the poorest to the second poorest quintile, such as in Burkina Faso (2003). In general, the NPD are typically between 50% and 75%. These results show that there are high risk children in all socioeconomic groups.

## 3.2 Quantifying within and between group variability

Fig 1 presents box plots showing the distribution of mortality risk for the last survey of each country. Countries are ordered from the highest median mortality risk (Sierra Leone) to the lowest median mortality risk (Ukraine). As the median mortality risk gets smaller, variance decreases as well. There is considerable overlap in mortality risk across countries. This suggests that country of birth explains only a small fraction of mortality risk and that all countries have some children with very high mortality risk.

Fig 2 presents the distribution of mortality risk across countries stratified by wealth quintile. Only the most recent survey is shown, and countries are ranked from highest to lowest median mortality risk, from top left to bottom right. Outliers are not shown and all graphs are presented on the same scale. For all countries and surveys in our sample, there is considerable overlap in mortality risk across socioeconomic groups within countries and this is true irrespective of a country's average mortality level. Among higher mortality countries, Sierra Leone and the Central African Republic have clear socioeconomic gradients in mortality risk. Among lower mortality countries, Bolivia, Brazil, Nigeria, and Cameroon have the largest socioeconomic gradients in mortality risk. High mortality countries like Niger and Lesotho exhibit no socioeconomic gradients in mortality, and this is also true for some lower mortality countries, such as Ukraine, Armenia and Jordan. Conclusions from Fig 2 are thus consistent with those from Table 1.

Table 2 presents results from our analysis. The first column gives the country and year in which the survey was taken, and first row presenting the results across all surveys combined. Columns two through five show the mean, median, and standard deviation of the mortality risk distribution from our analysis, and the $R^2$ of our ANOVA, which quantifies how much of the variance in mortality risk is explained by wealth quintile.

Globally, wealth quintile only explains about 3% of the variability in mortality risk. However, there is substantial country to country heterogeneity. The countries with the highest $R^2$ values are India (23%), Nigeria (17%), Indonesia (14%), and Cameroon (14%). In contrast, Eswatini, Lesotho, Tanzania, Moldova, Sao Tome and Principe, Kyrgyzstan, Uzbekistan, Kenya, Ukraine, and Comoros all have $R^2$ point estimates that are less than 1%. Further there is not a clear relationship between $R^2$ and mean/median mortality risk. Using country of birth in the ANOVA gives a posterior mean $R^2$ of 19%. Thus the ANOVA results confirm the findings from the boxplots of mortality rates in Figs 1 and 2 which show that while there is substantial country to country heterogeneity, within a given country wealth does not explain much of the variability in mortality risk.

Mortality risk distributions have a long right tail and in Table 2 the mean mortality risk is always higher than the median. In every country, there are individuals that face much higher mortality risk than the national average.

## 3.3 Comparing mortality among highest risk and poorest children

Poverty status alone is often used to decide which families will be targeted by health interventions. However, high within group variability for socioeconomic groups suggests that targeting

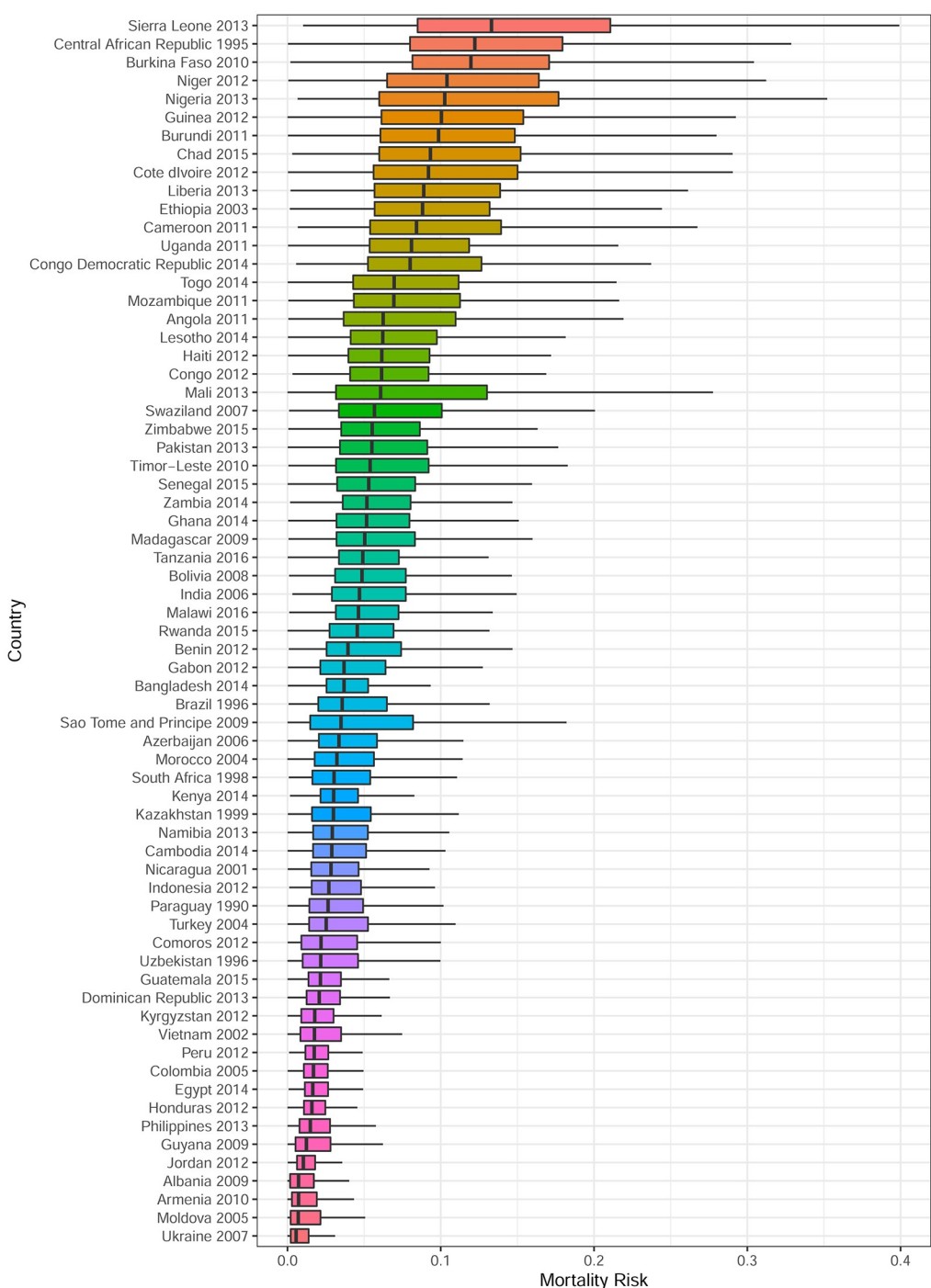

**Fig 1. Box plots for mortality risk by country and survey.** Lines are ±1.5 times interquartile range, boxes are lower to upper quartile, and dark line is the median mortality risk. Outliers are not shown.

based on a single demographic variable is inefficient because there are high risk births in all socioeconomic groups. We formally demonstrate the validity of this hypothesis for the last survey of each country, comparing efficiency gains of targeting the 20% poorest compared to targeting the 20% highest risk. Results are presented in Table 3. For all surveys and all countries,

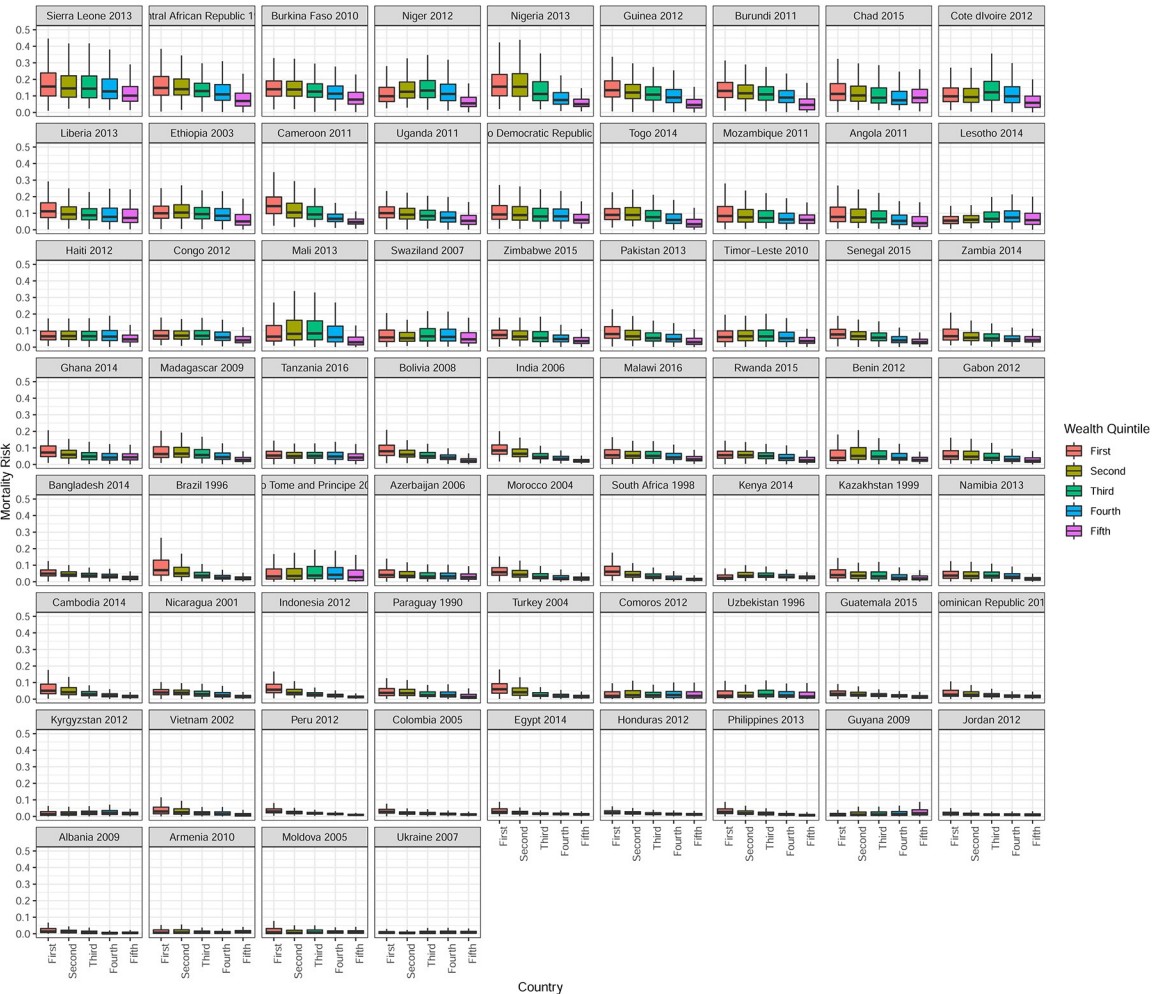

**Fig 2. Box plots for mortality risk by wealth quintile, country, and survey.** Countries are ranked from lowest to highest mortality. Lines are ±1.5 times interquartile range, boxes are lower to upper quartile, and dark line is the median mortality risk. Outliers are not shown.

our approach is much more efficient in identifying high risk births than targeting the poor. Efficiency gains range from 26% in Mali (1996), to more than 550% in Guyana (2009). Efficiency gains are not strongly related to a country's average mortality rates.

### 3.4 Who are the highest risk children?

We define the high risk (low risk) births for a particular country and survey as those in the top 20% (bottom 80%) of all births in terms of mortality risk as estimated by our model. For each of the continuous (categorical) variables, we calculate means of the variable for high and low risk births and the difference (odds ratio). Results are presented in Tables 1-7 in S1 Appendix for the last survey in each country. Higher risk births have younger mothers on average compared to lower risk births, but the differences are not substantively important: mothers from low risk group are usually less than a year older than mothers from the high risk group. High risk and low risk groups are also comparable for birth gender. For maternal education, there is often a significant difference between high risk and low risk births, but the difference is not substantively important. There is on average less than a year of additional education for

**Table 2. Results from ANOVA of posterior mean of mortality risk on wealth quintiles.** Countries are ordered by median mortality risk. Mean, median, variance, and $R^2$ are presented as posterior means and 95% intervals.

| Country | Mean | Median | Variance | $R^2$ (Wealth) |
|---|---|---|---|---|
| Overall | 6.9% (6.8%, 6.9%) | 4.0% (4.0%, 4.1%) | 0.7% (0.7%, 0.7%) | 3.3% (3.1%, 3.5%) |
| Sierra Leone 2013 | 15.7% (15.1%, 16.3%) | 12.7% (12.0%, 13.3%) | 1.3% (1.1%, 1.5%) | 2.8% (1.3%, 4.7%) |
| Central African Republic 1995 | 14.0% (13.0%, 15.1%) | 10.8% (9.8%, 12.0%) | 1.5% (1.2%, 1.8%) | 5.9% (2.7%, 10.3%) |
| Burkina Faso 2010 | 13.3% (12.8%, 13.8%) | 11.4% (10.8%, 11.9%) | 0.8% (0.7%, 0.9%) | 5.5% (3.2%, 8.3%) |
| Niger 2012 | 12.3% (11.7%, 12.8%) | 9.8% (9.2%, 10.4%) | 0.9% (0.8%, 1.1%) | 2.9% (1.3%, 4.9%) |
| Nigeria 2013 | 12.9% (12.6%, 13.3%) | 9.9% (9.5%, 10.3%) | 1.0% (0.9%, 1.1%) | 17.4% (14.9%, 20.0%) |
| Guinea 2012 | 11.6% (11.0%, 12.4%) | 9.1% (8.4%, 9.8%) | 1.0% (0.8%, 1.2%) | 7.7% (4.5%, 11.6%) |
| Burundi 2011 | 11.2% (10.4%, 12.0%) | 8.5% (7.7%, 9.3%) | 1.0% (0.8%, 1.3%) | 8.7% (4.9%, 13.1%) |
| Chad 2015 | 11.6% (11.2%, 12.1%) | 8.8% (8.4%, 9.3%) | 0.9% (0.8%, 1.0%) | 1.2% (0.3%, 2.3%) |
| Côte d'Ivoire 2012 | 11.5% (10.8%, 12.2%) | 8.2% (7.5%, 9.0%) | 1.1% (0.9%, 1.4%) | 1.4% (0.2%, 3.2%) |
| Liberia 2013 | 10.6% (9.9%, 11.3%) | 8.0% (7.4%, 8.6%) | 0.9% (0.7%, 1.0%) | 1.5% (0.2%, 3.6%) |
| Ethiopia 2003 | 10.2% (9.8%, 10.7%) | 8.0% (7.5%, 8.5%) | 0.7% (0.6%, 0.9%) | 3.6% (1.7%, 6.0%) |
| Cameroon 2011 | 10.7% (10.1%, 11.3%) | 7.8% (7.2%, 8.4%) | 0.9% (0.7%, 1.1%) | 13.6% (9.7%, 18.1%) |
| Uganda 2011 | 9.3% (8.7%, 9.9%) | 7.0% (6.4%, 7.7%) | 0.7% (0.6%, 0.9%) | 3.8% (1.5%, 6.9%) |
| Congo Democratic Republic 2014 | 9.9% (9.4%, 10.4%) | 7.5% (7.0%, 7.9%) | 0.7% (0.6%, 0.8%) | 2.5% (1.1%, 4.4%) |
| Togo 2014 | 8.7% (8.1%, 9.3%) | 6.1% (5.5%, 6.7%) | 0.8% (0.6%, 1.0%) | 5.5% (2.7%, 8.8%) |
| Mozambique 2011 | 8.9% (8.4%, 9.5%) | 6.1% (5.6%, 6.6%) | 0.8% (0.7%, 1.0%) | 2.2% (0.7%, 4.2%) |
| Angola 2011 | 8.8% (8.2%, 9.5%) | 5.4% (4.8%, 6.0%) | 1.0% (0.8%, 1.3%) | 2.4% (0.6%, 4.8%) |
| Lesotho 2014 | 8.1% (7.3%, 9.0%) | 4.6% (3.8%, 5.4%) | 1.2% (0.9%, 1.5%) | 0.4% (0.0%, 1.6%) |
| Haiti 2012 | 7.5% (6.9%, 8.0%) | 5.2% (4.7%, 5.8%) | 0.6% (0.5%, 0.8%) | 0.6% (0.0%, 1.9%) |
| Congo 2012 | 7.5% (7.0%, 8.1%) | 5.3% (4.8%, 5.8%) | 0.6% (0.5%, 0.7%) | 1.3% (0.1%, 3.1%) |
| Mali 2013 | 9.1% (8.6%, 9.7%) | 5.4% (4.9%, 6.0%) | 1.0% (0.9%, 1.2%) | 2.6% (1.2%, 4.3%) |
| Eswatini 2007 | 8.6% (7.6%, 9.6%) | 4.0% (3.3%, 4.9%) | 1.7% (1.3%, 2.1%) | 0.4% (0.0%, 1.5%) |
| Zimbabwe 2015 | 7.4% (6.8%, 8.0%) | 4.5% (4.0%, 5.1%) | 0.9% (0.7%, 1.1%) | 3.0% (1.2%, 5.4%) |
| Pakistan 2013 | 7.1% (6.7%, 7.6%) | 4.9% (4.5%, 5.4%) | 0.5% (0.4%, 0.6%) | 5.7% (3.2%, 8.7%) |
| Timor-Leste 2010 | 6.9% (6.4%, 7.4%) | 4.7% (4.2%, 5.1%) | 0.5% (0.4%, 0.6%) | 1.8% (0.5%, 3.9%) |
| Senegal 2015 | 6.3% (5.9%, 6.7%) | 4.8% (4.4%, 5.2%) | 0.3% (0.3%, 0.4%) | 7.6% (4.1%, 11.5%) |
| Zambia 2014 | 6.7% (6.3%, 7.1%) | 4.5% (4.1%, 4.9%) | 0.5% (0.4%, 0.6%) | 2.9% (1.2%, 5.1%) |
| Ghana 2014 | 6.4% (5.9%, 7.0%) | 4.1% (3.7%, 4.7%) | 0.5% (0.4%, 0.7%) | 2.8% (0.6%, 5.8%) |
| Madagascar 2009 | 6.7% (6.3%, 7.1%) | 4.5% (4.1%, 4.9%) | 0.5% (0.4%, 0.6%) | 6.0% (3.7%, 8.9%) |
| Tanzania 2016 | 6.0% (5.5%, 6.4%) | 4.1% (3.7%, 4.5%) | 0.4% (0.3%, 0.5%) | 0.3% (0.0%, 1.3%) |
| Bolivia 2008 | 6.1% (5.6%, 6.6%) | 4.1% (3.6%, 4.5%) | 0.5% (0.4%, 0.6%) | 11.2% (7.6%, 15.4%) |
| India 2006 | 5.9% (5.7%, 6.1%) | 4.3% (4.2%, 4.5%) | 0.3% (0.2%, 0.3%) | 22.7% (19.8%, 25.9%) |
| Malawi 2016 | 5.8% (5.5%, 6.2%) | 4.2% (3.9%, 4.5%) | 0.3% (0.2%, 0.4%) | 3.4% (1.6%, 5.7%) |
| Rwanda 2015 | 5.3% (4.8%, 5.8%) | 3.5% (3.1%, 4.0%) | 0.4% (0.3%, 0.5%) | 3.0% (0.9%, 6.0%) |
| Benin 2012 | 6.6% (6.2%, 7.0%) | 3.6% (3.2%, 3.9%) | 0.8% (0.6%, 0.9%) | 2.1% (1.1%, 3.4%) |
| Gabon 2012 | 5.5% (4.9%, 6.1%) | 2.7% (2.3%, 3.2%) | 0.7% (0.6%, 0.9%) | 2.3% (0.7%, 4.5%) |
| Bangladesh 2014 | 4.4% (4.1%, 4.7%) | 3.2% (2.9%, 3.5%) | 0.2% (0.2%, 0.3%) | 6.4% (3.4%, 9.8%) |
| Brazil 1996 | 5.6% (5.1%, 6.1%) | 2.7% (2.2%, 3.1%) | 0.8% (0.6%, 1.0%) | 9.9% (6.6%, 13.6%) |
| Sao Tome and Principe 2009 | 6.9% (6.0%, 8.0%) | 1.9% (1.3%, 2.6%) | 1.9% (1.4%, 2.3%) | 0.2% (0.0%, 1.2%) |
| Azerbaijan 2006 | 5.0% (4.3%, 5.8%) | 2.0% (1.5%, 2.7%) | 0.9% (0.6%, 1.1%) | 0.8% (0.0%, 2.4%) |
| Morocco 2004 | 4.6% (4.1%, 5.1%) | 2.4% (2.0%, 2.8%) | 0.5% (0.4%, 0.6%) | 6.0% (3.0%, 9.5%) |
| South Africa 1998 | 4.4% (3.9%, 4.9%) | 2.1% (1.7%, 2.5%) | 0.5% (0.4%, 0.7%) | 7.9% (4.7%, 11.8%) |
| Kenya 2014 | 4.0% (3.7%, 4.2%) | 2.6% (2.4%, 2.8%) | 0.2% (0.2%, 0.2%) | 0.1% (0.0%, 0.5%) |
| Kazakhstan 1999 | 4.5% (3.8%, 5.3%) | 1.8% (1.3%, 2.3%) | 0.8% (0.5%, 1.0%) | 1.7% (0.2%, 4.0%) |
| Namibia 2013 | 4.5% (4.0%, 5.0%) | 2.0% (1.6%, 2.4%) | 0.7% (0.5%, 0.9%) | 1.8% (0.5%, 3.8%) |

*(Continued)*

**Table 2.** (Continued)

| Country | Mean | Median | Variance | R² (Wealth) |
|---|---|---|---|---|
| Cambodia 2014 | 4.5% (4.1%, 5.0%) | 2.3% (2.0%, 2.6%) | 0.5% (0.4%, 0.6%) | 8.3% (5.5%, 11.6%) |
| Nicaragua 2001 | 3.6% (3.2%, 3.9%) | 2.1% (1.8%, 2.4%) | 0.2% (0.2%, 0.3%) | 4.2% (1.9%, 7.4%) |
| Indonesia 2012 | 4.0% (3.7%, 4.3%) | 2.2% (1.9%, 2.4%) | 0.3% (0.2%, 0.4%) | 13.7% (10.5%, 16.9%) |
| Paraguay 1990 | 4.1% (3.6%, 4.6%) | 1.8% (1.5%, 2.2%) | 0.5% (0.4%, 0.7%) | 2.0% (0.5%, 4.2%) |
| Turkey 2004 | 4.5% (4.0%, 5.1%) | 1.9% (1.5%, 2.3%) | 0.6% (0.5%, 0.8%) | 6.1% (3.3%, 9.7%) |
| Comoros 2012 | 4.1% (3.5%, 4.7%) | 1.1% (0.8%, 1.5%) | 0.8% (0.6%, 1.0%) | 0.1% (0.0%, 0.6%) |
| Uzbekistan 1996 | 4.4% (3.7%, 5.1%) | 1.3% (0.9%, 1.7%) | 0.9% (0.7%, 1.2%) | 0.1% (0.0%, 0.6%) |
| Guatemala 2015 | 2.9% (2.6%, 3.2%) | 1.7% (1.5%, 1.9%) | 0.2% (0.1%, 0.2%) | 4.7% (2.4%, 7.5%) |
| Dominican Republic 2013 | 3.2% (2.8%, 3.7%) | 1.3% (1.0%, 1.6%) | 0.4% (0.3%, 0.6%) | 2.5% (0.8%, 4.9%) |
| Kyrgyzstan 2012 | 2.8% (2.4%, 3.3%) | 0.9% (0.6%, 1.3%) | 0.5% (0.3%, 0.6%) | 0.1% (0.0%, 0.6%) |
| Vietnam 2002 | 3.0% (2.5%, 3.4%) | 1.0% (0.8%, 1.4%) | 0.4% (0.3%, 0.5%) | 2.1% (0.4%, 4.6%) |
| Peru 2012 | 2.3% (2.1%, 2.4%) | 1.5% (1.4%, 1.6%) | 0.1% (0.1%, 0.1%) | 10.5% (7.4%, 14.3%) |
| Colombia 2005 | 2.2% (2.0%, 2.4%) | 1.2% (1.0%, 1.4%) | 0.1% (0.1%, 0.2%) | 4.1% (2.1%, 6.6%) |
| Egypt 2014 | 2.5% (2.3%, 2.7%) | 1.3% (1.1%, 1.5%) | 0.2% (0.2%, 0.3%) | 3.5% (1.8%, 5.7%) |
| Honduras 2012 | 2.2% (2.0%, 2.5%) | 1.1% (0.9%, 1.3%) | 0.2% (0.1%, 0.3%) | 1.3% (0.3%, 2.8%) |
| Philippines 2013 | 2.5% (2.2%, 2.8%) | 0.9% (0.7%, 1.2%) | 0.3% (0.2%, 0.4%) | 3.8% (2.0%, 5.7%) |
| Guyana 2009 | 3.1% (2.5%, 3.7%) | 0.4% (0.1%, 0.7%) | 0.9% (0.7%, 1.3%) | 0.8% (0.1%, 1.9%) |
| Jordan 2012 | 1.7% (1.5%, 1.9%) | 0.7% (0.5%, 0.8%) | 0.2% (0.1%, 0.2%) | 0.5% (0.0%, 1.5%) |
| Albania 2009 | 2.4% (1.9%, 2.9%) | 0.1% (0.0%, 0.2%) | 0.9% (0.6%, 1.2%) | 1.0% (0.2%, 2.3%) |
| Armenia 2010 | 2.4% (1.9%, 3.1%) | 0.1% (0.0%, 0.3%) | 1.0% (0.7%, 1.3%) | 0.5% (0.0%, 1.6%) |
| Moldova 2005 | 2.9% (2.3%, 3.6%) | 0.0% (0.0%, 0.2%) | 1.2% (0.7%, 1.7%) | 0.3% (0.0%, 1.1%) |
| Ukraine 2007 | 1.8% (1.3%, 2.4%) | 0.0% (0.0%, 0.2%) | 0.7% (0.3%, 1.1%) | 0.1% (0.0%, 0.6%) |

mothers from the low risk group. There is also often a statistical, but not substantive difference in birth order.

The most substantial differences between the higher and lower risk groups are for residency (urban/rural), wealth, and previous death of a sibling. High risk births are substantively poorer than the remaining 80% of the population. In Cambodia, high risk births average at the poorest 32nd percentile of wealth while the low risk births average around the 53rd percentile of wealth. We find similar results for other countries: Bolivia: 32% against 52%; Brazil: 31% against 53%; Peru: 30% against 53%; Nigeria: 32% against 53%.

High risk births are disproportionately born to mothers that have already experienced a prior death of another child. The odds ratio is 18.8 (13.1, 26.7) in Benin; 16.3 (10.9, 24.1) in Mali; and 15.4 (11.9, 19.9) in Nigeria. Even for relatively wealthier countries, the odds ratio for another death is high for mothers that have experienced a prior death. The only countries in which a prior death is not a significant risk factor for a subsequent birth are Moldova and Vietnam. Ukraine seems an exception, but the fractions of the births with a prior death are small, and this makes the odds ratio for Ukraine not very meaningful.

## 4 Discussion

In this study we have investigated inequality in under-5 mortality within and between socioeconomic groups for a large pool of LMIC. We have made three related contributions to the existing research. First, we show that for all 67 countries in our sample, most of the variability in mortality risk exists within socioeconomic groups, not between groups. Second, we show that within countries the average mortality risk—which is closely related to national averages

**Table 3. Efficiency gains by targeting 20% highest risk as estimated from our model versus targeting the poorest 20%.** The first column gives the country and year. The second column gives sample size per survey. The third column is under 5 mortality rate in the 20% poorest. The fourth column is the mortality rate in the 20% identified as having the highest mortality risk for each sample with 95% posterior intervals. Efficiency Gain is defined as (HRDeaths—PoorDeaths)/PoorDeaths. CAR is Central African Republic.

| Country Year | Sample Size | Mortality Poor | Mortality High Risk | Efficiency Gains |
|---|---|---|---|---|
| Albania 2009 | 2,481 | 32% | 86% (74%, 94%) | 168% (132%, 195%) |
| Armenia 2000 | 2,602 | 22% | 53% (47%, 59%) | 144% (116%, 175%) |
| Armenia 2010 | 1,545 | 23% | 77% (63%, 91%) | 230% (170%, 290%) |
| Angola 2011 | 5,812 | 24% | 44% (41%, 46%) | 80% (69%, 89%) |
| Azerbaijan 2006 | 2,739 | 27% | 49% (42%, 55%) | 83% (59%, 107%) |
| Bangladesh 2000 | 9,061 | 26% | 39% (36%, 40%) | 50% (42%, 57%) |
| Bangladesh 2004 | 7,261 | 24% | 38% (35%, 40%) | 55% (44%, 65%) |
| Bangladesh 2007 | 6,929 | 25% | 43% (40%, 45%) | 69% (58%, 79%) |
| Bangladesh 2014 | 14,512 | 27% | 40% (37%, 42%) | 47% (39%, 56%) |
| Burkina Faso 1993 | 5,514 | 19% | 31% (29%, 33%) | 61% (51%, 70%) |
| Burkina Faso 1999 | 5,702 | 21% | 30% (28%, 31%) | 39% (32%, 46%) |
| Burkina Faso 2003 | 12,060 | 20% | 33% (32%, 34%) | 69% (62%, 74%) |
| Burkina Faso 2010 | 16,759 | 23% | 35% (33%, 36%) | 52% (47%, 57%) |
| Benin 1996 | 5,386 | 22% | 32% (31%, 34%) | 47% (39%, 55%) |
| Benin 2001 | 5,691 | 26% | 34% (32%, 36%) | 33% (26%, 41%) |
| Benin 2006 | 16,984 | 22% | 35% (34%, 36%) | 57% (52%, 63%) |
| Benin 2012 | 12,904 | 21% | 51% (49%, 53%) | 139% (129%, 148%) |
| Bolivia 1998 | 9,334 | 29% | 42% (40%, 45%) | 48% (40%, 55%) |
| Bolivia 2004 | 10,546 | 25% | 42% (40%, 44%) | 67% (58%, 75%) |
| Bolivia 2008 | 10,048 | 29% | 42% (40%, 45%) | 46% (37%, 55%) |
| Brazil 1996 | 6,023 | 34% | 51% (47%, 55%) | 50% (40%, 62%) |
| Burundi 2011 | 6,016 | 25% | 38% (35%, 40%) | 54% (45%, 64%) |
| Cambodia 2000 | 12,071 | 27% | 39% (38%, 41%) | 47% (42%, 53%) |
| Cambodia 2011 | 7,258 | 28% | 47% (43%, 49%) | 64% (53%, 74%) |
| Cambodia 2014 | 8,272 | 31% | 51% (48%, 54%) | 67% (56%, 77%) |
| CAR 1995 | 4,429 | 25% | 36% (34%, 39%) | 46% (35%, 54%) |
| Chad 1997 | 6,941 | 15% | 34% (33%, 36%) | 133% (123%, 143%) |
| Chad 2004 | 6,260 | 18% | 34% (33%, 36%) | 93% (83%, 102%) |
| Chad 2015 | 18,985 | 22% | 39% (38%, 40%) | 77% (72%, 82%) |
| Congo 2005 | 4,419 | 23% | 40% (38%, 42%) | 74% (63%, 85%) |
| Congo 2012 | 7,597 | 22% | 39% (36%, 41%) | 74% (63%, 84%) |
| Côte d'Ivoire 1999 | 2,757 | 25% | 42% (39%, 45%) | 73% (60%, 84%) |
| Côte d'Ivoire 2005 | 3,812 | 22% | 38% (35%, 41%) | 70% (56%, 83%) |
| Côte d'Ivoire 2012 | 7,224 | 21% | 40% (38%, 42%) | 94% (85%, 104%) |
| Cameroon 1991 | 3,140 | 26% | 41% (39%, 44%) | 57% (46%, 67%) |
| Cameroon 1998 | 4,080 | 30% | 43% (40%, 45%) | 43% (34%, 51%) |
| Cameroon 2004 | 7,535 | 26% | 40% (38%, 41%) | 52% (46%, 58%) |
| Cameroon 2011 | 10,812 | 29% | 39% (38%, 41%) | 35% (30%, 41%) |
| Colombia 1990 | 4,087 | 34% | 63% (56%, 70%) | 86% (65%, 105%) |
| Colombia 1995 | 5,041 | 30% | 56% (49%, 62%) | 90% (66%, 112%) |
| Colombia 2005 | 15,630 | 32% | 51% (46%, 57%) | 58% (43%, 76%) |
| Comoros 1996 | 2,208 | 22% | 42% (38%, 46%) | 93% (75%, 111%) |
| Comoros 2012 | 3,390 | 18% | 57% (51%, 62%) | 223% (190%, 253%) |
| DRC 2007 | 7,971 | 24% | 40% (39%, 42%) | 70% (64%, 76%) |
| DRC 2014 | 15,132 | 22% | 39% (37%, 40%) | 74% (68%, 81%) |

*(Continued)*

**Table 3.** (Continued)

| Country Year | Sample Size | Mortality Poor | Mortality High Risk | Efficiency Gains |
|---|---|---|---|---|
| DR 1999 | 3,250 | 26% | 50% (46%, 55%) | 97% (78%, 116%) |
| DR 2002 | 12,941 | 32% | 49% (45%, 52%) | 50% (40%, 61%) |
| DR 2007 | 13,945 | 26% | 46% (42%, 50%) | 77% (61%, 93%) |
| DR 2013 | 4,782 | 30% | 51% (45%, 56%) | 67% (48%, 85%) |
| Egypt 1996 | 12,791 | 30% | 41% (40%, 43%) | 40% (35%, 46%) |
| Egypt 2003 | 11,850 | 30% | 49% (46%, 52%) | 64% (54%, 72%) |
| Egypt 2008 | 11,394 | 32% | 51% (46%, 55%) | 59% (46%, 72%) |
| Egypt 2014 | 14,486 | 29% | 52% (48%, 56%) | 78% (64%, 91%) |
| Eswatini 2007 | 2,421 | 22% | 47% (42%, 51%) | 111% (91%, 129%) |
| Ethiopia 1997 | 12,984 | 18% | 38% (36%, 39%) | 109% (101%, 116%) |
| Ethiopia 2003 | 13,218 | 22% | 38% (37%, 40%) | 79% (71%, 86%) |
| Gabon 2001 | 3,783 | 20% | 43% (39%, 47%) | 111% (93%, 129%) |
| Gabon 2012 | 5,149 | 26% | 48% (44%, 52%) | 84% (70%, 98%) |
| Ghana 1994 | 3,281 | 24% | 42% (40%, 46%) | 77% (65%, 90%) |
| Ghana 1999 | 3,226 | 23% | 42% (39%, 45%) | 84% (69%, 98%) |
| Ghana 2003 | 4,134 | 25% | 41% (38%, 44%) | 63% (52%, 75%) |
| Ghana 2008 | 3,258 | 25% | 44% (40%, 49%) | 81% (64%, 99%) |
| Ghana 2014 | 6,370 | 27% | 41% (37%, 44%) | 53% (40%, 64%) |
| Guinea 1999 | 6,867 | 24% | 32% (30%, 34%) | 32% (26%, 39%) |
| Guinea 2005 | 7,807 | 22% | 33% (31%, 34%) | 48% (42%, 54%) |
| Guinea 2012 | 8,010 | 26% | 37% (35%, 39%) | 45% (38%, 52%) |
| Guatemala 1999 | 7,083 | 23% | 42% (38%, 45%) | 81% (68%, 95%) |
| Guatemala 2015 | 11,719 | 28% | 46% (43%, 49%) | 66% (54%, 79%) |
| Guyana 2005 | 1,268 | 14% | 86% (76%, 93%) | 525% (450%, 575%) |
| Guyana 2009 | 2,464 | 10% | 68% (59%, 77%) | 578% (489%, 667%) |
| Honduras 2006 | 12,380 | 23% | 43% (39%, 46%) | 88% (72%, 102%) |
| Honduras 2012 | 10,065 | 26% | 50% (44%, 55%) | 93% (71%, 113%) |
| Haiti 1995 | 3,020 | 24% | 40% (37%, 43%) | 65% (53%, 75%) |
| Haiti 2000 | 7,063 | 23% | 36% (34%, 38%) | 56% (48%, 64%) |
| Haiti 2006 | 5,907 | 26% | 41% (39%, 44%) | 60% (49%, 70%) |
| Haiti 2012 | 6,944 | 20% | 40% (37%, 43%) | 102% (88%, 116%) |
| India 1993 | 65,681 | 29% | 41% (41%, 42%) | 45% (42%, 47%) |
| India 2000 | 53,079 | 30% | 39% (38%, 39%) | 31% (29%, 33%) |
| India 2006 | 59,240 | 32% | 43% (42%, 44%) | 35% (32%, 38%) |
| Indonesia 1997 | 23,155 | 26% | 46% (45%, 48%) | 79% (73%, 85%) |
| Indonesia 2003 | 16,049 | 31% | 51% (48%, 53%) | 65% (57%, 73%) |
| Indonesia 2007 | 20,592 | 34% | 49% (48%, 51%) | 44% (39%, 49%) |
| Indonesia 2012 | 19,788 | 35% | 52% (49%, 54%) | 46% (39%, 53%) |
| Jordan 1990 | 9,308 | 28% | 47% (44%, 51%) | 67% (55%, 79%) |
| Jordan 1997 | 6,408 | 27% | 54% (50%, 59%) | 100% (84%, 119%) |
| Jordan 2002 | 7,098 | 23% | 48% (43%, 53%) | 108% (88%, 130%) |
| Jordan 2009 | 13,691 | 23% | 52% (47%, 56%) | 123% (104%, 141%) |
| Jordan 2012 | 11,205 | 26% | 54% (48%, 59%) | 106% (84%, 128%) |
| Kenya 1993 | 6,514 | 29% | 45% (42%, 48%) | 55% (46%, 65%) |
| Kenya 1998 | 5,789 | 28% | 51% (48%, 53%) | 83% (73%, 93%) |
| Kenya 2009 | 5,412 | 20% | 48% (45%, 51%) | 136% (120%, 150%) |
| Kenya 2014 | 23,924 | 16% | 44% (42%, 46%) | 179% (166%, 192%) |

(*Continued*)

**Table 3.** (Continued)

| Country Year | Sample Size | Mortality Poor | Mortality High Risk | Efficiency Gains |
|---|---|---|---|---|
| Kazakhstan 1999 | 2,651 | 25% | 50% (43%, 56%) | 103% (76%, 127%) |
| Kyrgyzstan 1997 | 2,400 | 27% | 52% (46%, 57%) | 92% (71%, 110%) |
| Kyrgyzstan 2012 | 3,705 | 19% | 53% (45%, 60%) | 184% (141%, 223%) |
| Liberia 2009 | 6,871 | 22% | 36% (35%, 38%) | 65% (57%, 73%) |
| Liberia 2013 | 8,220 | 24% | 37% (36%, 39%) | 59% (51%, 66%) |
| Lesotho 2005 | 3,115 | 23% | 46% (42%, 51%) | 103% (85%, 123%) |
| Lesotho 2010 | 3,107 | 17% | 45% (41%, 50%) | 160% (134%, 185%) |
| Lesotho 2014 | 3,250 | 17% | 44% (39%, 48%) | 158% (133%, 184%) |
| Morocco 1992 | 5,422 | 27% | 39% (36%, 42%) | 46% (35%, 57%) |
| Morocco 2004 | 6,493 | 30% | 49% (45%, 53%) | 63% (50%, 76%) |
| Moldova 2005 | 1,744 | 21% | 88% (72%, 96%) | 317% (244%, 358%) |
| Madagascar 1997 | 5,960 | 19% | 36% (34%, 38%) | 93% (83%, 103%) |
| Madagascar 2004 | 5,268 | 30% | 45% (42%, 48%) | 50% (40%, 59%) |
| Madagascar 2009 | 12,686 | 25% | 43% (41%, 45%) | 73% (64%, 81%) |
| Mali 1996 | 9,960 | 24% | 30% (29%, 31%) | 26% (22%, 30%) |
| Mali 2001 | 13,031 | 21% | 32% (31%, 33%) | 56% (52%, 59%) |
| Mali 2006 | 15,201 | 23% | 33% (32%, 34%) | 45% (41%, 49%) |
| Mali 2013 | 9,249 | 21% | 47% (45%, 48%) | 122% (113%, 131%) |
| Malawi 1992 | 4,746 | 21% | 34% (33%, 36%) | 62% (54%, 69%) |
| Malawi 2005 | 9,663 | 24% | 34% (32%, 35%) | 42% (35%, 48%) |
| Malawi 2010 | 20,677 | 22% | 34% (33%, 35%) | 53% (47%, 58%) |
| Malawi 2016 | 16,793 | 24% | 41% (39%, 43%) | 70% (61%, 77%) |
| Mozambique 1997 | 6,834 | 23% | 39% (38%, 41%) | 72% (65%, 78%) |
| Mozambique 2004 | 8,942 | 19% | 36% (34%, 37%) | 91% (83%, 99%) |
| Mozambique 2011 | 10,379 | 24% | 42% (40%, 44%) | 77% (69%, 85%) |
| Nicaragua 1998 | 8,665 | 20% | 46% (43%, 49%) | 131% (115%, 147%) |
| Nicaragua 2001 | 9,008 | 23% | 43% (40%, 47%) | 87% (72%, 103%) |
| Nigeria 1990 | 8,696 | 25% | 43% (42%, 45%) | 70% (65%, 75%) |
| Nigeria 2003 | 5,848 | 22% | 36% (34%, 37%) | 61% (54%, 67%) |
| Nigeria 2008 | 30,182 | 24% | 37% (36%, 37%) | 55% (52%, 58%) |
| Nigeria 2013 | 34,186 | 26% | 40% (39%, 41%) | 56% (53%, 58%) |
| Niger 1998 | 7,644 | 18% | 32% (31%, 33%) | 81% (75%, 86%) |
| Niger 2006 | 9,820 | 17% | 35% (33%, 36%) | 106% (98%, 114%) |
| Niger 2012 | 13,573 | 20% | 37% (36%, 39%) | 84% (77%, 89%) |
| Namibia 1992 | 3,692 | 22% | 46% (43%, 49%) | 104% (91%, 118%) |
| Namibia 2000 | 4,354 | 23% | 55% (50%, 59%) | 138% (119%, 157%) |
| Namibia 2007 | 4,668 | 29% | 50% (46%, 54%) | 73% (59%, 87%) |
| Namibia 2013 | 4,691 | 22% | 51% (46%, 56%) | 130% (108%, 152%) |
| Pakistan 1991 | 8,356 | 20% | 46% (45%, 48%) | 129% (120%, 138%) |
| Pakistan 2007 | 9,531 | 26% | 47% (44%, 49%) | 78% (69%, 87%) |
| Pakistan 2013 | 11,854 | 26% | 42% (40%, 44%) | 64% (56%, 71%) |
| Peru 1992 | 9,085 | 27% | 44% (42%, 46%) | 66% (58%, 74%) |
| Peru 1996 | 19,554 | 28% | 42% (40%, 44%) | 51% (45%, 58%) |
| Peru 2000 | 17,334 | 29% | 42% (40%, 44%) | 48% (41%, 54%) |
| Peru 2008 | 13,739 | 30% | 45% (41%, 49%) | 48% (35%, 60%) |
| Peru 2012 | 31,443 | 32% | 44% (41%, 46%) | 35% (27%, 42%) |

(*Continued*)

**Table 3.** (Continued)

| Country Year | Sample Size | Mortality Poor | Mortality High Risk | Efficiency Gains |
|---|---|---|---|---|
| Paraguay 1990 | 4,375 | 22% | 48% (43%, 53%) | 118% (96%, 139%) |
| Philippines 1993 | 9,340 | 27% | 48% (46%, 51%) | 78% (68%, 87%) |
| Philippines 1998 | 8,361 | 27% | 50% (46%, 53%) | 84% (72%, 97%) |
| Philippines 2003 | 7,863 | 33% | 53% (49%, 58%) | 63% (50%, 77%) |
| Philippines 2008 | 7,480 | 29% | 54% (50%, 58%) | 83% (70%, 97%) |
| Philippines 2013 | 8,159 | 32% | 56% (51%, 61%) | 77% (62%, 92%) |
| Rwanda 1992 | 6,071 | 18% | 37% (35%, 38%) | 99% (90%, 109%) |
| Rwanda 2005 | 9,139 | 22% | 34% (32%, 35%) | 52% (46%, 58%) |
| Rwanda 2008 | 4,865 | 18% | 43% (41%, 45%) | 143% (130%, 156%) |
| Rwanda 2015 | 8,096 | 24% | 41% (38%, 45%) | 74% (60%, 88%) |
| Sierra Leone 2008 | 6,413 | 23% | 40% (38%, 42%) | 71% (63%, 78%) |
| Sierra Leone 2013 | 13,981 | 22% | 36% (35%, 37%) | 64% (59%, 69%) |
| Senegal 1997 | 7,311 | 24% | 34% (32%, 36%) | 41% (33%, 49%) |
| Senegal 2005 | 10,284 | 26% | 36% (34%, 37%) | 36% (30%, 42%) |
| Senegal 2009 | 13,229 | 24% | 36% (34%, 37%) | 49% (43%, 55%) |
| Senegal 2015 | 12,606 | 27% | 38% (36%, 40%) | 45% (37%, 52%) |
| Sao Tome and Principe 2009 | 1,685 | 19% | 54% (48%, 60%) | 181% (150%, 212%) |
| Togo 1998 | 7,211 | 22% | 33% (32%, 35%) | 51% (43%, 58%) |
| Togo 2014 | 6,901 | 24% | 40% (38%, 43%) | 70% (60%, 80%) |
| Timor-Leste 2010 | 9,499 | 22% | 42% (39%, 44%) | 87% (76%, 97%) |
| Turkey 1993 | 4,998 | 33% | 49% (45%, 52%) | 47% (38%, 56%) |
| Turkey 1998 | 4,162 | 31% | 51% (46%, 56%) | 64% (48%, 78%) |
| Turkey 2004 | 4,765 | 30% | 51% (47%, 55%) | 68% (54%, 81%) |
| Tanzania 1999 | 6,715 | 20% | 36% (33%, 37%) | 81% (70%, 90%) |
| Tanzania 2005 | 7,200 | 23% | 36% (34%, 38%) | 55% (46%, 63%) |
| Tanzania 2010 | 11,262 | 25% | 38% (36%, 40%) | 54% (47%, 61%) |
| Tanzania 2016 | 8,745 | 21% | 40% (37%, 42%) | 86% (73%, 99%) |
| Ukraine 2007 | 1,494 | 19% | 87% (65%, 100%) | 350% (233%, 417%) |
| Uganda 1995 | 6,244 | 25% | 35% (33%, 37%) | 43% (35%, 51%) |
| Uganda 2001 | 5,933 | 25% | 36% (34%, 38%) | 46% (37%, 55%) |
| Uganda 2010 | 5,912 | 24% | 36% (34%, 38%) | 47% (38%, 56%) |
| Uganda 2011 | 7,852 | 24% | 37% (35%, 39%) | 57% (47%, 65%) |
| Uzbekistan 1996 | 2,656 | 24% | 55% (50%, 60%) | 129% (109%, 153%) |
| Vietnam 2002 | 4,060 | 29% | 54% (47%, 59%) | 83% (61%, 102%) |
| South Africa 1998 | 5,564 | 33% | 48% (44%, 53%) | 45% (31%, 58%) |
| Zambia 1997 | 5,614 | 23% | 35% (34%, 37%) | 52% (45%, 61%) |
| Zambia 2002 | 6,027 | 24% | 35% (33%, 37%) | 46% (39%, 55%) |
| Zambia 2007 | 5,808 | 16% | 36% (34%, 38%) | 122% (108%, 135%) |
| Zambia 2014 | 12,324 | 26% | 42% (39%, 44%) | 61% (51%, 69%) |
| Zimbabwe 1994 | 4,622 | 21% | 47% (43%, 50%) | 122% (103%, 141%) |
| Zimbabwe 1999 | 3,713 | 21% | 50% (46%, 54%) | 135% (115%, 153%) |
| Zimbabwe 2006 | 4,357 | 19% | 46% (41%, 50%) | 137% (112%, 162%) |
| Zimbabwe 2011 | 4,374 | 20% | 49% (45%, 54%) | 146% (124%, 168%) |
| Zimbabwe 2015 | 5,726 | 25% | 44% (41%, 46%) | 72% (61%, 82%) |

of child mortality—is far from the typical (modal) mortality risk experienced by most births. Third, we show that poverty status alone, while important, is a poor proxy for being at the higher risk of an an early death than the general population. All these findings have important policy implications. In addition, we have developed new methods to analyse inequality in mortality risk which have broad applicability.

While quantifying inequality in under-5 mortality between socioeconomic groups is important it misses a larger within-group inequality. In particular, we have shown that for most countries socioeconomic group explains less than 5% of the total variability in mortality. Even in countries where socioeconomic inequality matters the most, socioeconomic group explains very little of the variation in U5MR. For example, socioeconomic status explains 11% of U5MR in Bolivia and 22% in India. This means that there is a large overlap in mortality risk among births from different socioeconomic groups and, as a consequence, there is a large a number of high risk individuals outside that poorest group. In addition, being born to a particular country does not predict your mortality risk very well, which means that between country comparisons also miss most of the variability in mortality risk.

In addition of being incomplete, between country comparisons are often done in terms of average level of child mortality. However, we show that countries' distributions of mortality risk are right skewed because some births experience substantially higher mortality risk than the national averages. These are left behind populations who are largely unnoticed when we only look at average mortality in socioeconomic groups. The typical modal mortality rate in each country is very different from the national averages of child mortality. Thus between-country comparisons using national averages are not comparing typical mortality levels between countries.

Finally, most equity based policy strategies that target births are based on a single risk factor, usually poverty status. However, efficiency gains from targeting the 20% highest risk births versus the 20% poorest are substantively important for all countries that we have data for, with efficiency gains ranging from 26% in Mali (1996), to more than 550% in Guyana (2009), likely due to the fact that it is one of the few countries with an apparent decrease in mortality risk with increasing wealth. Although the 20% highest risk births are usually the poorest and from rural areas, as might be expected, including other risk factors and their interactions considerably improves the identification of left behind individuals.

One previously overlooked characteristic is the importance of having experienced a prior death of a child. [28, 29] This is likely the case because this variable represents several unmeasured risk factors at the maternal level. However, it is an observable variable and can be the object of policy targeting. And it should be used to do so. We find that this is a particularly important characteristic for Sub-Saharan Africa countries in our sample. For these countries, just targeting mothers that have already experienced the death of a child could be an effective way to reach high risk populations.

Taken together these results support the view that measuring national averages of under-5 mortality is insufficient to identify left behind groups. [5, 30–34] The concerns raised by United Nations General Assembly Resolution 68/261 are real and important, and we have shown that policy makers and international agencies should routinely implement disaggregation of inequality measures by several demographic variables simultaneously. [4] However, our findings suggest that monitoring inequality between socioeconomic groups of births may not enable policy makers to accurately identify many left behind children. We recommend using nationally representative surveys or administrative data to estimate mortality risk at the individual level to identify left behind populations that can be the target of interventions. We also recommend our methods to properly quantify and monitor high risk populations.

Our findings should not be interpreted as recommending against targeting the poor. Poverty *alone* is not the best guide for equity based policies because other risk factors are also important. Poverty status needs to be combined with other available information to identify high risk births. This is important for both low and high mortality countries, because children in need are spread out across socioeconomic groups. Further, since high risk children tend to be poor and from rural areas, most interventions that work for the poorest children will probably work for the highest risk children. Thus we are not suggesting major changes in interventions targeting high risk populations. Instead, we are proposing a new methodology that combines information from multiple well known risk factors simultaneously to identify high risk births. Our approach considers interactions among risk factors that are readily available for LMIC via nationally representative health surveys, and frees researchers and policy makers from having to decide which risk factors capture most of the inequality in each country-year.

The methods developed in this paper have broader applicability and are flexible enough to be applied to a number of different scenarios. For example, some countries with good vital registration system could use their administrative data instead of surveys. When people wish to implement an intervention in a particular country, our methodology points the way to a more targeted and impactful intervention. Implementers will need to choose variables, and they may choose different predictors than we have chosen, depending on data available and political and medical considerations. This is acceptable and something we consider a necessary part of implementing our methods in practice.

Our recommendations are also related to a large body of literature in medicine and public health that develops risk scores for individuals to identify those at risk of some event. These scores have been applied to a variety of outcomes and our results suggest the possible usefulness of such scores for identification of high risk children. [35] Our approach requires representative surveys of the population, such as DHS or Multiple Indicator Cluster Surveys (MICS) so that we can rank children by mortality risk based on demographics. Policy makers could use mobile apps, which are now widely used for data collection, to collect and combine information on the children, calculate their risk, and then check whether their score is above or below a pre-determined threshold. We would not suggest a single risk score for the entire world. Rather, we would develop a score for each country, and we would update the score as new data became available.

The calculus of the efficiency gains assumes that interventions have the same costs for each birth. In reality, costs need to be adjusted according to local conditions. However, our approach provides a baseline to which any other allocation algorithm should be compared. Every comparison allocation scheme also needs to accommodate costs, not just our allocation scheme. For example, targeting the poor is likely easier in urban settings than in rural settings, and this would be a differential cost for the simple "intervene with the poor" intervention. It is possible to incorporate costs; one would multiply estimated probability of mortality times cost, then follow our same procedure to identify a combination of cheapest and most at risk to intervene with, until the budget had been spent. Instead of identifying the 20% most at risk, one would tabulate costs until the allocation funds had been spent. No matter differential costs, combining information from multiple observable risk factors better identifies high risk populations. Having identified higher risk populations, public health officials can then work to bring down costs, and best target at-risk births.

Our methodology has not explicitly included the complex sampling design from the DHS. We did this to create a more parsimonious set of methodological innovations. We treated DHS samples as a random sample. However, we have included all variables used to stratify the surveys, which implicitly incorporates some of the sample design in our analysis. Future research should explicitly incorporate survey design.

In conclusion, our results show that despite progress toward reducing national averages of under-5 mortality, we still have substantial inequality within groups of births defined by commonly used stratifiers that measure progress toward SDG's. Our results suggest that researchers and policy makers should also quantify inequality in mortality risk within groups of births in addition to between-groups comparisons. Quantifying both between and within group inequality helps us to have an accurate picture of inequality in under-5 mortality and to identify left behind populations that otherwise cannot be easily identified.

## Supporting information

**S1 Appendix. [36].**
(PDF)

## Author Contributions

**Conceptualization:** Antonio P. Ramos, Robert E. Weiss.

**Data curation:** Antonio P. Ramos.

**Formal analysis:** Antonio P. Ramos, Martin J. Flores.

**Funding acquisition:** Antonio P. Ramos.

**Investigation:** Antonio P. Ramos.

**Methodology:** Antonio P. Ramos, Martin J. Flores, Robert E. Weiss.

**Project administration:** Antonio P. Ramos.

**Software:** Martin J. Flores.

**Supervision:** Robert E. Weiss.

**Writing – original draft:** Antonio P. Ramos, Martin J. Flores.

**Writing – review & editing:** Antonio P. Ramos, Martin J. Flores, Robert E. Weiss.

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
