## [Decision Letter · Decision Letter 0]

16 Oct 2019

PONE-D-19-25716

Leave No Child Behind:

Using Data from 4.5 Million Children from 77 Developing Countries to Measure Inequality Within and Between Socioeconomic Groups and to Identify Left Behind Populations.

PLOS ONE

Dear Dr Ramos,

Thank you for submitting your manuscript to PLOS ONE. After careful consideration, we feel that it has merit but does not fully meet PLOS ONE’s publication criteria as it currently stands. Therefore, we invite you to submit a revised version of the manuscript that addresses the points raised during the review process.

Two experts in the field provided a long list of comments, which called for more clarity on the choice of predictors, more details on some variables (e.g. location), more elaboration on the sensitivity of the results to other model specifications and programmatic implications of this proposed method for targeting high-risk children. The hypothesis that interventions have the same cost for each birth must be questioned, the cost associated with targeting by this new approach must be addressed. The interpretation of the variable "occurrence of a previous death" should also be revised as it alone summarizes exposure to all other predictors in the past and could introduce circularity into the reasoning. The implications of this variable for targeting at-risk populations need further discussion. Overall, the comments from these two reviewers should greatly help in building a more convincing paper.

We would appreciate receiving your revised manuscript by Nov 30 2019 11:59PM. To enhance the reproducibility of your results, we recommend that if applicable you deposit your laboratory protocols in protocols.io, where a protocol can be assigned its own identifier (DOI) such that it can be cited independently in the future. For instructions see: http://journals.plos.org/plosone/s/submission-guidelines#loc-laboratory-protocols

We look forward to receiving your revised manuscript.

Kind regards,

Bruno Masquelier, PhD

Academic Editor

PLOS ONE

Journal Requirements:

2. In your Methods section, please provide additional information about the survey included in this analysis. in particular, please describe the  criteria used for inclusion of surveys in your analysis, and specify whether any time limit was applied.

3. Our internal editors have looked over your manuscript and determined that it is within the scope of our Health Inequities and Disparities Research Call for Papers. This collection of papers is headed by a team of Guest Editors for PLOS ONE: Clare Bambra, Hans Bosma, Diana Burgess, Joseph Telfair, Barbara Turner, and Jennie Popay. The Collection will encompass a diverse range of research articles on health inequities and disparities.  Additional information can be found on our announcement page: https://collections.plos.org/s/health-inequities.

If you would like your manuscript to be considered for this collection, please let us know in your cover letter and we will ensure that your paper is treated as if you were responding to this call. If you would prefer to remove your manuscript from collection consideration, please specify this in the cover letter.

Reviewers' comments:

Reviewer's Responses to Questions

**Comments to the Author**

1. Is the manuscript technically sound, and do the data support the conclusions?

Reviewer #1: Yes

Reviewer #2: No

2. Has the statistical analysis been performed appropriately and rigorously? 

Reviewer #1: I Don't Know

Reviewer #2: Yes

3. Have the authors made all data underlying the findings in their manuscript fully available?

Reviewer #1: Yes

Reviewer #2: No

4. Is the manuscript presented in an intelligible fashion and written in standard English?

Reviewer #1: Yes

Reviewer #2: Yes

5. Review Comments to the Author

Reviewer #1: 1. I find the idea of identifying vulnerable groups of children using multiple markers at a time quite interesting. The objective of the exercise, however, has to be very clear so that the right set of predictors are selected.

2. On p. 7, the authors say they used predictors commonly present in SDG monitoring studies. The list is familiar, except the mother experiencing a previous child death.

3. The authors state “The range of the wealth variable varies from survey to survey. Therefore in each survey we transform the original wealth variable to the fraction of households that have equal or lower wealth than the current wealth value.” I think this needs clarification. I can’t really understand what was done here.

4. Following this phrase, the authors say birth order was included. It is not clear if it was included in the wealth index or as a further predictor in the model.

5. The authors mention location as a variable, but they do not explain how location was defined or used in the models.

6. Which are the continuous variables? Most of the predictors used in monitoring exercises are categorized, and it is clearly stated which variables were categorized and which were modeled in their continuous form.

7. I am not familiar with these Bayesian models, so I will not comment on the strategies for model fitting. A statistical reviewer familiar with the method is needed.

8. On p. 8 the authors state “Under the assumption that intervention has the same cost for each birth, we calculate the efficiency gain in targeting the highest risk births…”. This assumption clearly does not hold, or every health program would start focusing on the most vulnerable groups. These groups are usually much more difficult to reach due to a series of factors including distance to health services, inability to pay for the services or even cover the cost of transportation, culture, and so on.

9. What the authors present in Table 1 is called under-five mortality rate. Child mortality rate is the probability of dying for children 1-4 years. The authors do not describe in methods how the U5MR was calculated by wealth quintile.

10. What the authors tried to summarize using the proportion of non-poor deaths in better estimated using the concentration index. See papers by Adam Wagstaff for details.

11. The authors focus on the 20% poorest as if this was the only group at higher risk, but as they recognize, in most countries there is a gradient. And it is quite obvious that mortality will happen in all quintiles.

12. The caption for Table 1 needs to be carefully revised.

13. In p. 10 the authors say that there are high risk children in all socioeconomic groups. It seems to be a individual level assertion on risk, and it is not clear how it can be derived from the results presented so far.

14. In p. 11 the authors seem to imply that only a few countries present a gradient in mortality relative to wealth. But it is nearly impossible, especially for lower mortality countries, to infer that visually from Fig. 2 given the scale. I suggest the authors choose some measure for wealth related trend in mortality.

15. In p. 12-13 the results of the modeling are compared to the mortality of the groups defined by each predictor separately. Not surprisingly poverty and living in a rural area are the ones that best identify the high-risk children. Except, of course, for a previous death. This is a variable that is very hard to interpret in the context of this analysis since it summarizes in itself exposure earlier in time for all of the other potential risk factors.

16. I think that the selection of predictors for the model needs a more thorough explanation for its rationale. In my opinion it mixes up social determinants with biological determinants that have very different meanings in terms of designing policies. You can focus policies, at population level, on geographic areas, on poor families, and so on. History of a previous infant death can only be used for individual level targeting at a health service and with people that are already in contact with the service. In summary, these aspects should be better explained, justified and discussed in the paper.

17. Finally, a comment on the focus of between group vs within group variation. This is a well known phenomenon and in the vast majority of cases individual level variation will far exceed group level variation.

Reviewer #2: This paper deals with an important topic that is actual and relevant within the SDG framework. It aims to improve strategies for identification of births of highest mortality risks, thus providing a better tool for targeting this group. While relevant, I have some concerns about the methodology and some of the conclusions.

1. The authors used a selected set of equity stratifiers and characteristics to predict the risk of child mortality among children under-five born 5-10 years before the surveys, using a Bayesian hierarchical logistic model. They then extracted the top 20% of births with highest predicted mortality risk of mortality, which are referred to as high risk birth. They compared this group to the births in the bottom quintile of the wealth score, and conclude that their model provides a better way to identify the top 20% of births with highest mortality risk than the bottom 20% wealth quintile. This argument is trivial due to the very fact that their model include the wealth index, in addition to several other stratifiers. Furthermore, the two groups are not necessarily comparable. The high risk group of births is predicted from a mortality model while the bottom quintile is based on an independently defined socio-economic group. There has been no discussion on the model specification and prediction. The predictions obtained are dependent on the variables included in the model, and clearly a different specification may lead to a different group of high risk births. I think they are comparing apples and oranges.

2. The authors defined an efficiency measure by computing the relative difference in child mortality between the two groups. The only assumption that was stated was that cost of interventions is assumed the same for all births. However, they are completely silent on differences in the cost of identifying and targeting each group. The high risk group that they came up with is much more difficult and potentially more costly to identify and target than groups based on socio-economic and demographic characteristics.

3. The authors assumes wrongly that the bottom quintile is often the target for equity-based policy. Equity policies always involves several other stratifiers, including gender, place of residence, level of education, etc. The SDGs actually recommend disaggregation of indicators across many of these stratifiers. However, what has not been often done is cross-disaggregation by multiple stratifiers to identify the highest risk groups. I think this is where the multivariate model that the author propose might have some value but it is also important to highlight its complexity for policy and program translation.

4. While the authors claim that their approach provides an improved way to identify left behind group of children, the recommendation for identifying this groups is less clear and therefore less amenable to policy actions, while simplistic approaches using separately the equity dimensions provides a more tangible and actionable policy actions because the groups identified can be easily targeted.

5. I also found it a bit flaw to assume that perfect equity is realized when the distribution of deaths between the 20% and the top 80% wealth quintiles is also 20% and 80% respectively. This is because the bottom quintile usually has higher fertility and therefore proportionally higher number of births that the upper quintiles. The authors themselves acknowledge this fact in the methods but went on the compare the proportion of non-poor deaths (NPD) to 80%. What is missing is the proportional distribution of births across the quintiles.

6. The statistical analysis and interpretation may need some revision. For e.g. boxplots are used and compared across countries and groups to make conclusions as if statistics tests were done. To conclude that country of birth explains only a small fraction of mortality risk should not just be based on eyeballing boxplots (page 11 paragraph 1). Although the boxplots overlap between Sierra Leone and Ukraine, it cannot be assume that births face the same risks in these two countries. Similarly boxplots on figure 2 were described in the same way (page 11).

7. It is interesting that the authors made an effort to characterize the highest risk children. However, the description in this section (page 12) is hard to follow. This because the summaries described are not reported in the tables referred to (table 3-9). Furthermore, the findings were counter-intuitive. For example maternal education is a major factor of differential mortality in children. Same for birth order (especially first born children face high mortality risk) and maternal age. This should be explain and discussed further. It is however not surprising that prior experience of death came out as a strong predictor of high risk. This is a bit of a circular reasoning: groups who experience death leave in places of high mortality.

8. Page 11, second paragraph: it is puzzling that Nigeria and Cameroon are referred to as lower mortality countries. this is not what figure 1 suggests.

9. Births in the 5-10 years preceding the survey were used in the analysis to control mortality exposure. It is unclear to me whether the 4,585,342 births were related to this period or refer to the entire births in the datasets. In addition it is indicated that more than one survey in each country was used to track trends over time (page 9 last paragraph) but I didn’t see any trend analysis in the paper.

10. Table 1: results for Pakistan look strange.

11. The text needs some editing in several places

6. PLOS authors have the option to publish the peer review history of their article (what does this mean?). If published, this will include your full peer review and any attached files.

Reviewer #1: No

Reviewer #2: No

---

## [Author Response · Author response to Decision Letter 0]

3 Dec 2019

Review Comments to the Author

Please use the space provided to explain your answers to the questions above. You may also include additional comments for the author, including concerns about dual publication, research ethics, or publication ethics. (Please upload your review as an attachment if it exceeds 20,000 characters). 

Reviewer #1: 

1. I find the idea of identifying vulnerable groups of children using multiple markers at a time quite interesting. The objective of the exercise, however, has to be very clear so that the right set of predictors are selected.

Thanks for your comment. Indeed, we believe that the choice of the predictors is of foremost importance. In our case, because we were interested in comparing a large set of countries and we were concerned about the Sustainable Development Goals (SDG), we use stratifiers commonly used to quantify inequality in the SDG to estimate mortality risk. (https://www.equidade.org/indicators). Our approach generalizes current approaches that evaluate progress toward the SDG by either (a) looking at each one of these stratifiers separately or (b) combines only a few of the stratifiers. We use all of them plus interactions. 

In addition, we also included two other well-known risk observables: (a) geographic location and (b) prior death.

2. On p. 7, the authors say they used predictors commonly present in SDG monitoring studies. The list is familiar, except the mother experiencing a previous child death.

Thanks for your comment. We included prior death because we want to include a variable that could aid program targeting and summarize a number of risk factors that were not included in the data collection. We find that births from a mother that had already experienced a prior death of a child to be at a very high risk of death. Policy makers should pay special attention to these births. We now explain the definition of this 0-1 indicator variable more carefully on page 7-8, including emphasizing that for a first birth the indicator is necessarily zero. 

3. The authors state “The range of the wealth variable varies from survey to survey. Therefore in each survey we transform the original wealth variable to the fraction of households that have equal or lower wealth than the current wealth value.” I think this needs clarification. I can’t really understand what was done here.

Thanks for the opportunity to clarify this point. The range of the continuous version of wealth variable varies from survey to survey, as it does not have a natural metric. Thus the wealth variable needs to be standardized somehow to be comparable across surveys. Several studies use wealth quintiles, which are comparable. However, using quantiles make a rich source of information poorer, because it makes a continuous variable into a categorical variable (quintiles). Thus we decided to use the cumulative distribution function (cdf) of the wealth variable, which is continuous, bonded between (0,1) and has a similar interpretation as quintiles, and is comparable across surveys. In particular, the cdf is the fraction of people who have less wealth than the mother in question. We have rephrased the definition more carefully on appendix A1. 

4. Following this phrase, the authors say birth order was included. It is not clear if it was included in the wealth index or as a further predictor in the model.

Birth order was included in the model as an independent predictor. We now clarify the definition on page 8, at the end of the first paragraph.

5. The authors mention location as a variable, but they do not explain how location was defined or used in the models.

Location is ‘sampling cluster’, the smallest geographic unit available from the DHS surveys that is comparable across all surveys. We treat sampling cluster as a random effect; births from the same sampling cluster are more similar than births from different sampling clusters. 

6. Which are the continuous variables? Most of the predictors used in monitoring exercises are categorized, and it is clearly stated which variables were categorized and which were modeled in their continuous form.

Whenever possible, we have try to use continuous variables. Continuous variables are birth order, maternal age, cumulative distribution function of wealth, maternal education (in years). Categorical: location (sampling cluster), gender, place of residency (urban or rural). 

7. I am not familiar with these Bayesian models, so I will not comment on the strategies for model fitting. A statistical reviewer familiar with the method is needed.

Thanks for your comment. We have answered a number of questions regarding our models for the second reviewer. We hope we have clarified our methodology. 

8. On p. 8 the authors state “Under the assumption that intervention has the same cost for each birth, we calculate the efficiency gain in targeting the highest risk births…”. This assumption clearly does not hold, or every health program would start focusing on the most vulnerable groups. These groups are usually much more difficult to reach due to a series of factors including distance to health services, inability to pay for the services or even cover the cost of transportation, culture, and so on.

Thank you for the comment. We know that this assumption does not hold. However, it provides a baseline to which any other allocation algorithm would be compared. Also, every allocation scheme also needs to accommodate costs, not just our allocation scheme. For example, targeting the poor is likely easier in urban settings than in rural settings, and this would be a differential cost for the simple 'intervene with the poor' intervention. Costs can be incorporated; For example, one would multiply estimated probability of mortality times cost, then follow our same procedure to identify a combination of cheapest and most at risk to intervene with, until the budget had been spent. Instead of identifying the 20% most at risk, one would tabulate costs until the allocation funds had been spent. We now include discussion about this limitation in the discussion section of the paper, pp. 17. No matter differential costs, our broader point remains: Combining information from multiple observable risk factors better identifies high risk populations. Having identified higher risk populations, public health officials can then work to bring down costs, and best target at-risk births. 

9. What the authors present in Table 1 is called under-five mortality rate. Child mortality rate is the probability of dying for children 1-4 years. The authors do not describe in methods how the U5MR was calculated by wealth quintile.

Thanks, you, we will clarify this in the text. We are indeed talking about under-5 mortality (U5MR). In each wealth quintile, U5MR was calculated by counting the proportion of deaths among all births in the respective quintile. We now say this on page 10, section 3.1. 

10. What the authors tried to summarize using the proportion of non-poor deaths in better estimated using the concentration index. See papers by Adam Wagstaff for details.

Thank you for the references. We are quite aware of concentration index, and it is indeed useful, particularly for cognoscenti. However, we are attempting to keep our calculations simpler for researchers of all backgrounds and we find the proportion of non-poor deaths useful and much easier to explain. Thus we prefer to calculate the proportion of non-poor deaths directly instead of using the concentration index.

11. The authors focus on the 20% poorest as if this was the only group at higher risk, but as they recognize, in most countries there is a gradient. And it is quite obvious that mortality will happen in all quintiles.

Yes, we agree that mortality will happen at all quintiles, and this is indeed a primary point of our article. Another key point is that this is happening at a much higher level than we were expecting to see. A third point is that targeting tends to only target the lowest socio-economic groups, to the detriment of high risk births in other wealth quintiles. We have augmented our original analysis by including an ANOVA analysis that that formally quantifies how much of the variance in mortality risk can be explained by membership to a particular socioeconomic group. 

12. The caption for Table 1 needs to be carefully revised.

Thank you for pointing this out. We have revised the caption. It now says: 

“Detailed description of data. N is the survey sample size used in our analysis. U5MR is the under-5 mortality rates by age five for each survey. Non-poor deaths (NPD) is the fraction of deaths from the top 80\\%. DRC is Democratic Republic of Congo, DR is Dominican Republic, and CAR is Central African Republic. The first quintile is the poorest births and the firth quantile is the richest births.”

13. In p. 10 the authors say that there are high risk children in all socioeconomic groups. It seems to be a individual level assertion on risk, and it is not clear how it can be derived from the results presented so far.

Yes, this is an assertion about individual level risk. We have estimated mortality risk at the individual level, for each birth in our data set using all the predictors and interactions we detailed in section 2.2. Thus we were able to estimate risk for individual births and identify the risk belonging to births, from make such an assertion. 

14. In p. 11 the authors seem to imply that only a few countries present a gradient in mortality relative to wealth. But it is nearly impossible, especially for lower mortality countries, to infer that visually from Fig. 2 given the scale. I suggest the authors choose some measure for wealth related trend in mortality.

Thank you for your comment. We have fixed the text so that now it is clearer. There is a gradient, and it is visible in figure 2. As the reviewer notes, it is harder to see the gradient for lower mortality countries; we have inspected the graph at higher resolution and the gradients are visible; also the gradient is discernable in table 2. 

15. In p. 12-13 the results of the modeling are compared to the mortality of the groups defined by each predictor separately. Not surprisingly poverty and living in a rural area are the ones that best identify the high-risk children. Except, of course, for a previous death. This is a variable that is very hard to interpret in the context of this analysis since it summarizes in itself exposure earlier in time for all of the other potential risk factors.

Thank for your question. We believe this is an important point that needs clarification.

This variable is only a forward looking variable: it measures prior death, not any death; thus future deaths are not used to predict past mortality. It does summarize several risk factors that were not included in the analysis. That is precisely why we think it is useful. Policy makers should particularly consider births from a mother that already experience the prior death of a child to be at higher risk of mortality and therefore be in need of intervention. 

16. I think that the selection of predictors for the model needs a more thorough explanation for its rationale. In my opinion it mixes up social determinants with biological determinants that have very different meanings in terms of designing policies. You can focus policies, at population level, on geographic areas, on poor families, and so on. History of a previous infant death can only be used for individual level targeting at a health service and with people that are already in contact with the service. In summary, these aspects should be better explained, justified and discussed in the paper.

Thanks for bringing up this point. We have now explained better sections 2.1 and 2.2 why we chose the variables we chose. In particular, variables such as mother’s education and mother’s age are mother-specific, in addition to prior infant death as the reviewer notes. These variables all have the potential to be trackable, to be used to identify births in need of intervention. Some variables may be easier to use to identify births at risk, some may be harder. We have illustrated how to identify high-risk births with our particular choice of variables, but in practice in any given locality and any given situation, more or fewer variables may need to be used for the full model used to estimate risk. We now say this in the discussion section on page 17. 

17. Finally, a comment on the focus of between group vs within group variation. This is a well known phenomenon and in the vast majority of cases individual level variation will far exceed group level variation.

Yes, we agree. However, we were surprised by how little the group membership explained the total variance in mortality risk. The new version of the paper includes an ANOVA that quantifies how much of the variation in mortality risk is within versus much between group. 

Reviewer #2: This paper deals with an important topic that is actual and relevant within the SDG framework. It aims to improve strategies for identification of births of highest mortality risks, thus providing a better tool for targeting this group. While relevant, I have some concerns about the methodology and some of the conclusions.

1. The authors used a selected set of equity stratifiers and characteristics to predict the risk of child mortality among children under-five born 5-10 years before the surveys, using a Bayesian hierarchical logistic model. They then extracted the top 20% of births with highest predicted mortality risk of mortality, which are referred to as high risk birth. They compared this group to the births in the bottom quintile of the wealth score, and conclude that their model provides a better way to identify the top 20% of births with highest mortality risk than the bottom 20% wealth quintile. This argument is trivial due to the very fact that their model include the wealth index, in addition to several other stratifiers. Furthermore, the two groups are not necessarily comparable. The high risk group of births is predicted from a mortality model while the bottom quintile is based on an independently defined socio-economic group. There has been no discussion on the model specification and prediction. The predictions obtained are dependent on the variables included in the model, and clearly a different specification may lead to a different group of high risk births. I think they are comparing apples and oranges.

Thank you for this comment. The model is discussed on section 2.2 of the main paper and, as space considerations required we omit some details, more fully in the appendix A2. With respect, we disagree, we are not comparing apples and oranges. We use novel methodology to make two different but related points: (1) most of the inequality exists within socioeconomic groups, not between them; (2) high risk births exist across all socioeconomic groups. Ignoring these facts while attempting to implement interventions to reduce under-five mortality will lead to inefficient targeting and for monitoring the SDG, ignoring this will lead to incomplete inequality monitoring. The comparison is entirely apt as state of the art for implementing interventions typically relies on use of one or a limited set of covariates to identify births worth intervening on. 

Our results are, of course, dependent on (a) the statistical model used to estimate mortality risk and (b) the included variables. In fact, (a) and (b) are also true for any other similar analysis. However, we believe that our results are less dependent on (a) and (b) than previous ones. 

To deal with (a) we develop a very flexible statistical model, that barely assumes any pre-defined functional form. For example, we have not done any model selection. Instead, we fit the most flexible possible model we could with available software and covariates for all countries. For each included variable we have their main effects and up to their 4-way interactions. Continuous variables were modeled via splines. Our priors avoid in-sample overfitting and promote regularization by imposing increasing constraints on higher order interactions. 

To deal with (b) we included all available stratifiers from the DHS that were comparable among a large set of countries and their interactions: https://www.equidade.org/indicators. In addition, we use two well known and observable and risk factors for early-life mortality: location and prior death (more on the later below). By including these risk factors that are common to other similar studies, we promote the comparability of our results. 

We would expect more accurate predictions by combining information from several risk factors. That is precisely why we developed our methodology. We see our approach as a generalization of the current approaches that quantify inequality by comparing one or a few stratifiers. In fact, our methods were developed to respond to the suggestion from the SDG that health indicators should be stratified by demographic variables when necessary, as we discuss in our introduction. However, the SDG suggestion leaves open a number of questions. For example, which stratifiers are important in each country? What about interactions? Our methods offer an answer to these questions. 

When people wish to implement an intervention in a particular country, our methodology points the way to a more targeted and impactful intervention. Implementers will need to choose variables, and they may choose different predictors than we have chosen, depending on data available and political and medical considerations. This is acceptable and something we consider a necessary part of the practical issues for implementing our methods in practice. 

2. The authors defined an efficiency measure by computing the relative difference in child mortality between the two groups. The only assumption that was stated was that cost of interventions is assumed the same for all births. However, they are completely silent on differences in the cost of identifying and targeting each group. The high risk group that they came up with is much more difficult and potentially more costly to identify and target than groups based on socio-economic and demographic characteristics.

Thank you for your comment. It is true that a more complex algorithm may have increased costs . As we have discussed in one of our responses to reviewer one, we know that this assumption does not hold. However, it provides a baseline to which any other allocation algorithm would be compared. Also, every allocation scheme also needs to accommodate costs, not just our allocation scheme. For example, targeting the poor is likely easier in urban settings than in rural settings, and this would be a differential cost for the simple 'intervene with the poor' intervention. Costs can be incorporated; For example, one would multiply estimated probability of mortality times cost, then follow our same procedure to identify a combination of cheapest and most at risk to intervene with, until the budget had been spent. Instead of identifying the 20% most at risk, one would tabulate costs until the allocation funds had been spent. Our computation can be implemented, for example, in an app, so canvassers could enter numbers and perform a calculation in real time. Alternatively, computer data bases are becoming more common even in LMIC countries, and the algorithm could be run at the clinic level for example. Exact details of implementation will depend on the ground truth in the country at the implementation stage of an intervention. 

We now include discussion about this limitation in the discussion section of the paper, pp. 17. No matter differential costs, our broader point remains: Combining information from multiple observable risk factors better identifies high risk populations. Having identified higher risk populations, public health officials can then work to bring down costs, and best target at-risk births. 

3. The authors assumes wrongly that the bottom quintile is often the target for equity-based policy. Equity policies always involves several other stratifiers, including gender, place of residence, level of education, etc. The SDGs actually recommend disaggregation of indicators across many of these stratifiers. However, what has not been often done is cross-disaggregation by multiple stratifiers to identify the highest risk groups. I think this is where the multivariate model that the author propose might have some value but it is also important to highlight its complexity for policy and program translation.

Thank you for this. First it is true that several interventions use a single risk factor, most commonly wealth/income/ poverty status, to target births at risk. We cite many of them in the introduction to our paper. For example, that is the case with cash transfer programs, which are now widely implemented in the world, as we discuss in the text on page 5. 

Second, yes, we know that sometimes targeting involves other stratifiers, usually only a few them. However, we have not seen any methodology that flexibly uses several stratifiers and their interactions simultaneously. 

Third, as we discussed in the introduction of the paper, we are also aware that the SDG suggest that, where relevant, inequality monitoring and program targeting should be disaggregated by several risk factors. Indeed, their suggest is the very motivation for our paper, as we discuss in the first paragraph of our paper. 

4. While the authors claim that their approach provides an improved way to identify left behind group of children, the recommendation for identifying this groups is less clear and therefore less amenable to policy actions, while simplistic approaches using separately the equity dimensions provides a more tangible and actionable policy actions because the groups identified can be easily targeted.

While we acknowledge that our approach can make targeting more complex than simpler approaches, we believe that the costs associated with our approach are largely offset by its benefits. In addition, we can do such a much better job of identifying those who need to be targeted, even if current government policies cannot really target all of the high risk people, it is still important to know how to target those at greatest risk, even if government can not currently implement this perfectly. 

5. I also found it a bit flaw to assume that perfect equity is realized when the distribution of deaths between the 20% and the top 80% wealth quintiles is also 20% and 80% respectively. This is because the bottom quintile usually has higher fertility and therefore proportionally higher number of births that the upper quintiles. The authors themselves acknowledge this fact in the methods but went on the compare the proportion of non-poor deaths (NPD) to 80%. What is missing is the proportional distribution of births across the quintiles.

Thanks for this comment: this is an important point that needs clarification. When we refer to the deaths among the wealthiest 80%, we mean80% of births and not the 80% of households. So for example if the poorest 10% of households were responsible for 20% of all births, we would consider non poor deaths as those deaths that occurred among the wealthiest 90% of households, not the wealthiest 80% of households. Thus we have an almost exact proportional distribution of 20% of births in each wealth quintile. We now clarify this in the by adding a new appendix explaining how our version of the wealth index was calculated. 

6. The statistical analysis and interpretation may need some revision. For e.g. boxplots are used and compared across countries and groups to make conclusions as if statistics tests were done. To conclude that country of birth explains only a small fraction of mortality risk should not just be based on eyeballing boxplots (page 11 paragraph 1). Although the boxplots overlap between Sierra Leone and Ukraine, it cannot be assume that births face the same risks in these two countries. Similarly boxplots on figure 2 were described in the same way (page 11). 

Box plots are used (a) to summarize the results of our analyses, (b) to illustrate the variation in estimated mortality risk across countries and (c) as adjuncts to formal tests. We have augmented our visualization analysis with a formal variance decomposition technique (explained in the appendix) that quantifies the within and between-group variances. Results of these are now included in Table 2. As we expected, it supports our earlier claims based on box plots. 

7. It is interesting that the authors made an effort to characterize the highest risk children. However, the description in this section (page 12) is hard to follow. This because the summaries described are not reported in the tables referred to (table 3-9). Furthermore, the findings were counter-intuitive. For example maternal education is a major factor of differential mortality in children. Same for birth order (especially first born children face high mortality risk) and maternal age. This should be explain and discussed further. It is however not surprising that prior experience of death came out as a strong predictor of high risk. This is a bit of a circular reasoning: groups who experience death leave in places of high mortality.

All predictors included in the model are important and well-known risk factors. They are also correlated with each other. Due to space constrains, we were unable to discuss all of them in great depth. 

Prior death identifies children that are at greater mortality risk than other children. It does summarize a wide variety of potential risk factors that were not included in the original data collection. It does aid program targeting because policy makers can direct their target efforts toward these children. It is not a ‘circular’ variable because it is a forward looking variable: For each birth, it only uses information about births that have already happened, and does not use information from the future. 

8. Page 11, second paragraph: it is puzzling that Nigeria and Cameroon are referred to as lower mortality countries. this is not what figure 1 suggests.

Thanks for pointing this out. We have fixed this in the text. 

9. Births in the 5-10 years preceding the survey were used in the analysis to control mortality exposure. It is unclear to me whether the 4,585,342 births were related to this period or refer to the entire births in the datasets. In addition it is indicated that more than one survey in each country was used to track trends over time (page 9 last paragraph) but I didn’t see any trend analysis in the paper.

Thanks for your comment. The 4,585,342 births were born during the 5-10 years preceding each survey and these births are all used in one or another of our formal analyses. We clarify this in the text at the end of the first paragraph on pp. 7. 

10. Table 1: results for Pakistan look strange.

Due to small sample size, we’re unable to calculate under-5 mortality rates for Pakistan in 1991. We have deleted this from our collection of analyses and do not report on this particular survey in the revision. 

11. The text needs some editing in several places.

We have edited the document thoroughly several times now and we hope it is better now.

---

## [Decision Letter · Decision Letter 1]

13 Feb 2020

PONE-D-19-25716R1

Leave No Child Behind:

Using Data from 4.5 Million Children from 77 Developing Countries to Measure Inequality Within and Between Groups of Births and to Identify Left Behind Populations

PLOS ONE

Dear Dr Ramos,

Many thanks for revising your manuscript entitled “Using Data from 4.5 Million Children from 77 Developing Countries to Measure Inequality Within and Between Socioeconomic Groups and to Identify Left Behind Populations”, to respond to the comments from the reviewers. My impression is that you addressed most of the reviewers' concerns, and one reviewer involved in the first round agreed to review the paper again and also acknowledged that most comments have been addressed. Yet, after reading again your paper in detail, I decided to send it to a third reviewer, who provided constructive and useful comments, and asked some questions of clarification that I would like you to consider in order to further improve the manuscript (see the attachment file). This revision should then lead to acceptance.  I also provide a few comments below. 

We would appreciate receiving your revised manuscript by Mar 29 2020 11:59PM. To enhance the reproducibility of your results, we recommend that if applicable you deposit your laboratory protocols in protocols.io, where a protocol can be assigned its own identifier (DOI) such that it can be cited independently in the future. For instructions see: http://journals.plos.org/plosone/s/submission-guidelines#loc-laboratory-protocols

We look forward to receiving your revised manuscript.

Kind regards,

Bruno Masquelier, PhD

Academic Editor

PLOS ONE

Additional Editor Comments (if provided): 

- You do not seem to account for age of the child in the model, and as a result, the estimates presented in Table 1 should not be interpreted as under-five mortality rates, these are not comparable to estimates reported in DHS reports or UN IGME values (which are period 5Q0). Your estimates, if I understand correctly the methodology, are estimates referring to a cohort of children born between 5 and 10 years before the survey. This should be highlighted in the manuscript (on page 10?). Also note that Table 1 uses CMR for under-five mortality, which is a bit confusing since it is often used for child mortality, from age 1 to 5. Could you change to U5MR (from birth to age 5)? There is no need to specify under-five mortality rates "by age 5", since by definition U5MR is the risk of a newborn dying before age 5. 

- The fact that U5MR sometimes increases from the lowest to the second quintile is also observed with estimates from DHS, but this inscrease is not necessarily significant. Would it be possible to add confidence intervals in Table 1?

- You indicate that you transformed the original wealth index but refer the readers to the appendix. Could you add one sentence in the main text about this transformation to explain how your index differs from the standard DHS index?

- Figure 1 presents box plots showing the distribution of mortality risk. To what unit of observation do these risks correspond? To each child? Or is this a distribution of predicted proportions of children deceased before age five for each combination of the covariates? This is unclear. You state that variability is correlated with median mortality levels. How is the variability estimated here? On an absolute scale? In that case, it is logical that the variability is greater in high mortality countries. Please explain.

A few typos: 

1) in the abstract (interpretation), differences do not explain

2) In the caption of Table 1, fifth quantile instead of firth quantile, and I suggest mentioning "births from the richest households" and "births from the poorest households" instead of referring to rich and poor births.

3) Brurundi on page 27

Some minor comments:

1) on page 7, you indicate that "We analyze under-5 mortality and thus we exclude births that did not occur at least five years prior to the survey." Under-five mortality is sometimes estimated for periods up to the survey, by assuming that mortality rates of the older age groups are kept constant or decline at a certain rate, hence suggest removing "thus" in this sentence.

Reviewers' comments:

Reviewer's Responses to Questions

**Comments to the Author**

1. If the authors have adequately addressed your comments raised in a previous round of review and you feel that this manuscript is now acceptable for publication, you may indicate that here to bypass the “Comments to the Author” section, enter your conflict of interest statement in the “Confidential to Editor” section, and submit your "Accept" recommendation.

Reviewer #1: All comments have been addressed

Reviewer #3: (No Response)

2. Is the manuscript technically sound, and do the data support the conclusions?

Reviewer #1: Yes

Reviewer #3: Yes

3. Has the statistical analysis been performed appropriately and rigorously? 

Reviewer #1: Yes

Reviewer #3: Yes

4. Have the authors made all data underlying the findings in their manuscript fully available?

Reviewer #1: Yes

Reviewer #3: Yes

5. Is the manuscript presented in an intelligible fashion and written in standard English?

Reviewer #1: Yes

Reviewer #3: Yes

6. Review Comments to the Author

Reviewer #1: (No Response)

Reviewer #3: (No Response)

7. PLOS authors have the option to publish the peer review history of their article (what does this mean?). If published, this will include your full peer review and any attached files.

Reviewer #1: Yes: Professor Aluisio J D Barros

Reviewer #3: Yes: Jessica Godwin

---

## [Author Response · Author response to Decision Letter 1]

8 Aug 2020

Please, see our file "response_to_reviewers_PlosOne _2" for detailed responses.

---

## [Editor Report · Decision Letter 2]

26 Aug 2020

Leave No Child Behind: Using Data from 1.7 Million Children from 67 Developing Countries to Measure Inequality Within and Between Groups of Births and to Identify Left Behind Populations

PONE-D-19-25716R2

Dear Dr. Ramos,

We’re pleased to inform you that your manuscript has been judged scientifically suitable for publication and will be formally accepted for publication once it meets all outstanding technical requirements.

Kind regards,

Bruno Masquelier, PhD

Academic Editor

PLOS ONE

Additional Editor Comments (optional):

Please take advantage of the final edits to correct some typos:

- Page 9 - there seems to be a typo in the formula for the equity gain as the ratio is multiplied twice by 100.

- Page 11- please revise "If the poorest 20% contain more than its share of deaths," and "variancedecreases".

- Page 12 - Swaziland is now Eswatini

- Page 15- an early death instead on early-death

- Page 17 - to reach high risk populations instead of to reach a high risk populations.

- Page 18 - below instead of bellow

- Page 19 - "Costs are possible to be incorporated;" should be revised (e.g. It is possible to incorporate costs)

- Tables and Figures: please replace Swaziland with Eswatini and Cote dIvoire by Côte d'Ivoire. Please Capitalize the first letter of headings in Table 3.
---

## [Editor Report · Acceptance letter]

15 Sep 2020

PONE-D-19-25716R2 

Leave No Child Behind: Using Data from 1.7 Million Children from 67 Developing Countries to Measure Inequality Within and Between Groups of Births and to Identify Left Behind Populations 

Dear Dr. Ramos:

I'm pleased to inform you that your manuscript has been deemed suitable for publication in PLOS ONE. Congratulations! Your manuscript is now with our production department. 

Kind regards, 

on behalf of

Dr. Bruno Masquelier 

%CORR_ED_EDITOR_ROLE%

PLOS ONE